# Validation of an interpretable data-driven wake model using lidar measurements from a field wake steering experiment

Balthazar Arnoldus Maria Sengers[1,2], Gerald Steinfeld[1], Paul Hulsman[1], and Martin Kühn[1]

[1]ForWind, Carl von Ossietzky Universität Oldenburg, Institute of Physics, Küpkersweg 70, 26129 Oldenburg, Germany
[2]Current affiliation: Fraunhofer IWES, Küpkersweg 70, 26129 Oldenburg, Germany

**Correspondence:** Balthazar Sengers (balthazar.sengers@uni-oldenburg.de)

**Abstract.** Data-driven wake models have recently shown a high accuracy in reproducing wake characteristics from numerical data sets. This study used wake measurements from a lidar-equipped commercial wind turbine and inflow measurements from a nearby met mast to validate an interpretable data-driven surrogate wake model. The trained data-driven model was then compared to a state-of-the-art analytical wake model. A multi-plane lidar measurement strategy captured the occurrence of the wake curl during yaw misalignment, which had not yet conclusively been observed in the field. The comparison between the wake models showed that the available power estimations of a virtual turbine situated four rotor diameters downstream were significantly more accurate with the data-driven model than with the analytical model. The Mean Absolute Percentage Error was reduced by 19 % to 36 %, depending on the input variables used. Especially under turbine yaw misalignment and high vertical shear, the data-driven model performed better. Further analysis suggested that the accuracy of the data-driven model is hardly affected when using only supervisory control and data acquisition (SCADA) data as input. Although the results are only obtained for a single turbine type, downstream distance and range of yaw misalignments, the outcome of this study is believed to demonstrate the potential of data-driven wake models.

## 1 Introduction

With the wind energy industry maturing, more focus is put on maximizing the power yield of existing assets. This involves moving away from the traditional, and currently still standard, greedy control of individual turbines to an optimization on wind farm level. In recent years, especially the wake steering concept has received considerable attention in the literature, in which the turbine is intentionally misaligned with the inflow wind, introducing a lateral component of the thrust force that deflects the wake away from a downstream turbine. Many aspects of this strategy have been studied over the years, including the underlying physics (e.g., Howland et al., 2016; Bastankhah and Porté-Agel, 2016) and its characteristics under different atmospheric conditions (e.g., Vollmer et al., 2016; Schottler et al., 2017). Additionally, the implementation of this concept in the field with so-called yaw controllers has received attention. Such controllers typically include a representation of the wake in the form of engineering wake models used to solve the optimization problem, as well as the design of the yaw controller itself (e.g., wind direction robustness (Rott et al., 2018; Simley et al., 2020), hysteresis (Kanev, 2020) and open- versus closed-loop (Doekemeijer et al., 2020; Howland et al., 2020)).

Although a large body of knowledge about the wake steering concept has been obtained, the industry appears to be hesitant to adapt due to the large uncertainties and lack of validation (van Wingerden et al., 2020; Boccolini et al., 2021). One limitation is the number of field experiments carried out. Due to the considerable expense and inaccessibility of test turbines, most research groups revert to high-fidelity simulations or wind tunnel experiments. Although they provide higher degree of reproducibility and more flexibility in choosing the studied scenarios, these experiments take place in controlled environments and do not

fully represent the complexity of the field. Wake models and yaw controllers are consequently developed based on data from idealized conditions. Their accuracy in field situations is questionable due to limited validation, slowing down the adoption by industry. This uncertainty is amplified by findings that the application of wake steering can lead to power losses under certain conditions (e.g., Fleming et al., 2020; Doekemeijer et al., 2021).

Several field campaigns have been conducted in recent years to study wake steering control. In their pioneering work, Wagenaar

et al. (2012) used a scaled wind farm to demonstrate the concept. Using rear-facing nacelle-mounted lidars, asymmetries in wake deflection depending on the sign of the yaw angle were observed for the near (Trujillo et al., 2016) and far wake (Bromm et al., 2018). This asymmetry is also found using numerical tools (e.g., Fleming et al., 2015) and attributed to shear-induced initial wake deflection (Gebraad et al., 2016) or the Coriolis force (Archer and Vasel-Be-Hagh, 2019). One prominent aspect associated with wake steering is the development of the wake curl as observed in numerical and wind tunnel experiments (e.g.,

Howland et al., 2016; Vollmer et al., 2016; Hulsman et al., 2022b). Fleming et al. (2017a) included a short notion that a curled shape could be observed in the field, while Brugger et al. (2020) did not find a curled wake in their field experiment. They argued that the effect of wind veer was too large for the counter-rotating vortices to generate a curled wake, with wind veer reported to tilt the wake in one direction (Herges et al., 2017; Brugger et al., 2019).

Using fixed yaw misalignment angles, Howland et al. (2019) found statistically significant gains of up to 47 % for low wind

speeds and a certain wind direction in a small wind farm consisting of six turbines. Ahmad et al. (2019) reported that wake steering is mainly beneficial in partial wake situations. Fleming et al. (2021) found an asymmetry of the downstream turbine power generation, where gains from correct steering (wake steered away from turbine) are larger than the losses from erroneous steering (wake steered into turbine). They attributed this effect to the added wake recovery induced by the counter-rotating vortices that also generate the wake curl.

Additionally, several controller test studies have been carried out, in which instead of a fixed yaw angle, an optimal yaw angle is employed based on the inflow conditions. This optimal yaw angle is determined with low-fidelity wake models which generate discretized look-up tables (LUTs). In a series of papers from the National Renewable Energy Laboratory (NREL), different versions of the FLOw Redirection and Induction in Steady State (FLORIS, NREL (2022)) framework have been used to generate these LUTs. In a field campaign at an offshore wind farm with a turbine spacing of 7 to 8 rotor diameters, Fleming

et al. (2017b) reported a 10 % power gain for certain wind directions. Fleming et al. (2019, 2020) showed results of a field-test with closely spaced turbines with two different versions of FLORIS, both resulting in a power gain for most conditions, but clear power losses for some wind directions. Lastly, Doekemeijer et al. (2021) found large power gains of up to 35 % for one wind direction sector with a two-turbine setup in complex terrain, but also here large losses were found for other wind directions.

These studies are pivotal in demonstrating the potential of wake steering, but also indicate that there is a large variability in its demonstrated effectiveness. Next to atmospheric inflow conditions, this can be attributed to turbine type, turbine spacing and terrain. Additionally, the choice of yaw controller and accuracy of the wake model used to develop the LUTs are believed to have an effect.

After the pioneering wake deficit models of Jensen (1983) and Ainslie (1988), Jimenez et al. (2010) first came up with a wake deflection model under yaw misalignment. Nowadays, most analytical wake models are based on the Gaussian model (Bastankhah and Porté-Agel, 2014, 2016; Niayifar and Porté-Agel, 2016). Combined with the curl wake model (Martínez-Tossas et al., 2019), the Gaussian-Curl Hybrid (GCH) model (King et al., 2021) prescribes the effect of counter-rotating vortices generated by turbine yaw misalignment, such as yaw-induced wake recovery, asymmetric deflection, and secondary steering. Lastly, Bastankhah et al. (2022) presented an analytical way to describe the development of the wake curl with downstream distance, and Bay et al. (2022) tackled "deep array" effects, in which many wakes interact deep inside a large wind farm, with the cumulative-curl model.

In addition to these analytical models, data-driven wake (surrogate) models have received some attention in recent years. Most use complex neural networks (e.g. Ti et al., 2020; Renganathan et al., 2022; Purohit et al., 2022; Asmuth and Korb, 2022) and have shown highly accurate results. However, these models need lots of training data and have an extremely low interpretability (black-box). In an attempt to overcome this, Sengers et al. (2022) presented an interpretable Data-driven wAke steeRing surrogaTe model (DART). Using only linear equations, DART uses inflow and turbine variables to estimate wake parameters such as deficit, center location and curl. It has a reduced number of parameters and is therefore highly interpretable and needs fewer training data. In a comparison using large eddy simulation (LES) results, Sengers et al. (2022) demonstrated that DART outperformed the Gaussian and GCH models, especially under stable atmospheric conditions.

As mentioned before, studies validating wake models with field measurements are rare, especially when yaw misalignments are included, resulting in uncertainties about their accuracy. Moreover, comparisons between analytical and data-driven models in their abilities to reproduce the characteristics of wakes observed in the field is done sporadically. However, validations with measurements and comparisons between models are necessary to assess their performance and provide direction for future work.

The objective of this paper is to use nacelle-based lidar measurements of the wake of a commercial turbine to validate the DART model and compare its accuracy with that of the GCH model. To achieve the objective, this study comprises of three components: (1) To design a scanning strategy able to capture wake characteristics such as deficit, center position and curl to accurately reconstruct a vertical cross-section of the wake, (2) to assess the performance of the wake models by their ability to estimate the available power of a virtual downstream turbine observed by the lidar, (3) to investigate DART's performance as function of data set size and input variables, including an analysis whether the model could operate on supervisory control and data acquisition (SCADA) data alone.

## 2 Measurement campaign

This section introduces the field experiment carried out within this study. Section 2.1 describes the measurement site and general setup. Section 2.2 describes the yaw control experiment. Sections 2.3 through 2.7 then discuss the devices, their measurement strategies and data processing. Especially in Sect. 2.3 more details are provided, including results from a preliminary study to determine the scanning strategy of the nacelle lidar, since the measurements from this device are essential for this study. Lastly, Sect. 2.8 describes how the data from all devices are used to select 10-minute averaged cases considered in the rest of the study.

### 2.1 Measurement site

Measurements were carried out in the period of February through April 2021 as a part of a yaw-control field campaign at a slightly hilly onshore site in north-eastern Germany located approximately 13.5 km from the Baltic sea, see also Hulsman et al. (2022a). The layout of the site, including the positioning of the measurement equipment, is shown in Fig. 1. The nacelle of turbine T1 was equipped with a downstream facing *Leosphere Windcube 200S* (serial no. WLS200S-024) pulsed lidar (Sect. 2.3). T1 was a commercial 3.5 MW *eno126* turbine with a hub height of 117 m and a rotor diameter $D$ of 126 m. The nacelle was further equipped with a *Thies Clima* wind vane and cup anemometer (Sect. 2.6), as well as a *Trimbl SPS* three-antenna GNSS system (hereafter called GPS) to measure orientation, tilt and roll (Sect. 2.7). A second pulsed lidar of the same type (serial no. WLS200S-023) was installed west of the turbine to measure inflow profiles (Sect. 2.4). North of this turbine, a meteorological mast (MM, Sect. 2.5) was erected and equipped with *Thies Clima* cup anemometers and wind vanes.

Lastly, Fig. 1 shows that a small 6 m high hill $5D$ upstream of T1 and a larger 27 m high hill $8D$ downstream of T1 were exactly in the wind direction sector that was not used due to the presence of the downstream turbines (see Sect. 2.2). Two villages with low buildings were located about 1 km from T1, directly upstream for wind directions around $\delta = 265°$ and $\delta = 320°$, mainly outside of the studied wind direction sectors. The dominant vegetation in the area is of agricultural nature, with patches of trees and bushes between the fields. These trees could affect the measurements for $\delta \approx 350°$, as noted in Hulsman et al. (2022a) using data from the same site. This influence was accepted, as omitting this sector would result in large data losses.

### 2.2 Yaw control experiment

As these measurements were part of a larger field campaign, only the wind direction sector $\delta = [268°, \ 360°] \cup [0°, \ 20°]$ could be used for experiments for this study. Unfortunately, in this sector two smaller turbines (T3 and T4) were located downstream of the lidar-equipped turbine. For the objectives of this study, measurements at $4D$ downstream were targeted. This was to avoid the near wake, as the two investigated wake models fail to represent the non-Gaussian shape of the wake deficit, and to ensure that the wake curl had developed. The wind speed reduction due to the induction zone of T3 at $4.8D$ (hub height of 103 m and a diameter of 93 m) was estimated to be in the order of 2 % (estimated with the vortex sheet theory (Medici et al., 2011)) at the targeted distance of $4D$. Although not ideal, no alternative was possible due to the restrictions of the measurement site and it was decided to neglect the effects of this induction zone.

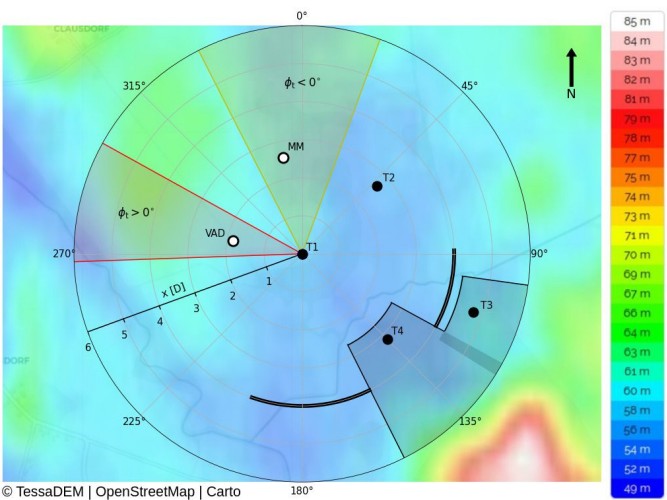

**Figure 1.** Layout of the measurement site with the local topography, relative to mean sea level, indicated in the background. Black markers indicate turbines, where T1 is equipped with the nacelle lidar. White markers indicate the met mast (MM) and ground-based lidar (VAD). Shaded areas indicate the wind direction sector with $\phi_t > 0°$ (red) and $\phi_t < 0°$ (yellow) and where wake measurements are assumed to be disturbed by the downstream turbines (grey). Thick black solid line indicates the measured locations used for analysis. (Source topographic map including color bar: topographic-map.com (2022)).

Part of the wind direction sector could not be used due to the positioning of T4 at $3.2D$ downstream. To make sure that the wake was not steered into T4, in the sector $\delta = [268°,\ 316°]$ the turbine toggled between target yaw misalignment angles of $\phi_t = 0°$ (duration of 30 minutes) and $\phi_t = +15°$ (duration of 60 minutes, clockwise rotation looking from above), steering the wake to the left. Correspondingly, in the sector $\delta = [316°,\ 360°] \cup [0°,\ 20°]$ the turbine toggled between $\phi_t = 0°$ (30 minutes) and $\phi_t = -15°$ (60 minutes, counterclockwise rotation looking from above), steering the wake to the right. The downside of this approach was that directly comparing positive and negative yaw angles under similar atmospheric conditions was not possible. Additionally, more data was collected in the first sector as this wind direction was more dominant.

Fixed yaw offsets were applied as this involved minimal changes to the yaw controller. Besides, a distribution of yaw misalignments was expected to be obtained due to the imperfect tracking of the wind direction by the yaw controller.

### 2.3 Nacelle lidar

This section describes the measurements performed with the nacelle-mounted lidar. Section 2.3.1 describes the design of the scanning strategy, including results of a numerical evaluation to determine what trajectory should be implemented in the field. Section 2.3.2 describes the processing, including filtering, of this data.

### 2.3.1 Design scanning strategy

A pulsed lidar can be mounted onto the nacelle to sample to turbine's wake. When operated with a single plan position indicator (PPI) scan with an elevation angle of $\phi_{\mathrm{PPI}} = 0°$, the line-of-sight velocities on a horizontal plane at hub height are obtained. Although quick, this trajectory only provides data at one height in the wake. Attempts have been made to capture information in the vertical plane, such as in Beck and Kühn (2019) who proposed a scanning pattern of alternating PPI and range height indicator (RHI) scans to obtain information in both dimensions. However, wake shape deformations due to wind veer (tilted) or yaw misalignment (curled) cannot be captured with this scanning strategy. Brugger et al. (2019, 2020) used nine PPI scans at different elevation angles, allowing to describe non-circular wake shapes in a vertical plane.

In this paper, their strategy was adopted and evaluated numerically to gain insights on how the number of PPI scans and their angular speed (following Carbajo Fuertes and Porté-Agel (2018)) affect the ability to capture the characteristics of 10-minute averaged wake. This exercise used large eddy simulation (LES) results, allowing for a systematic uncertainty analysis of the proposed scanning patterns.

The PArallelized Large-eddy-simulation Model (PALM, Maronga et al. (2020)) coupled with the aeroelastic code FAST (Jonkman and Buhl Jr., 2005; Krüger et al., 2022) representing the NREL 5MW turbine (Jonkman et al., 2009) provides the numerical wind fields. Precursor simulations generated realistic inflow conditions, after which main simulations with one turbine were performed. The aeroelastic code for the turbine installed in the field, as used in Sect. 4.1, was not yet available during the planning stage of this campaign. Both turbine T1 and the NREL 5MW turbine have the same rotor diameter (126 m), but differ in hub height (117 vs 90 m) and aeroelastic properties. It was, however, assumed that at $4D$ the characteristics of the wakes produced by these turbines are sufficiently similar.

A single turbine with yaw angles of $\phi = (-15°,\ 0°,\ 15°)$ in a neutral (TI = 10.3 %, $\alpha = 0.17$) and a stable (TI = 5.7 % and $\alpha = 0.32$) boundary layer with a hub height wind speed $U_{\mathrm{h}} \approx 8\ \mathrm{m\ s^{-1}}$ was simulated. The simulation length was 25 minutes, of which the first 15 minutes were omitted as spin-up and the remaining 10 minutes were used for analysis. Synthetic lidar data targeting $4D$ downstream were subsequently generated by employing the lidar simulator LiXim (Trabucchi, 2019) with an accumulation time of 0.1 s and an opening angle of 70°. Temporal averages were taken for all points in the scanning cycle. The wake composition method, later described in Sect. 3.1, was used to reconstruct vertical cross-sections of the wake, from which the available power $P_{\mathrm{av}}$ could be determined. This estimate is compared to the reference ($_{\mathrm{ref}}$) 10-minute averaged LES data. Used as metric is the Absolute Percentage Error (APE) over the six (two boundary layers times three yaw angles) simulations calculated with Eq. (1):

$$\mathrm{APE}\ [\%] = \left| \frac{P_{\mathrm{av}} - P_{\mathrm{av,ref}}}{P_{\mathrm{av,ref}}} \right| \cdot 100 \tag{1}$$

in which $P_{\mathrm{av}} = P/C_{\mathrm{P}} = 0.5\ \rho\ A\ U_{\mathrm{eq}}$ with $\rho$ the air density (assumed to be constant), $A$ the rotor area and $U_{\mathrm{eq}}$ the rotor equivalent wind speed. The bar 'all' on the far left in Fig. 2 indicates the reconstruction of the wake based on the original LES data, hence the error introduced by the composition method. Further, 1, 3, 5, 7 and 9 PPI scans were tested, where the middle scans always targeted hub height and the outermost scans upper and lower tip height at $4D$. Trajectories with an even number of PPI scans were not tested, as this would remove the scan at hub height that was needed for another study. Additionally, it is

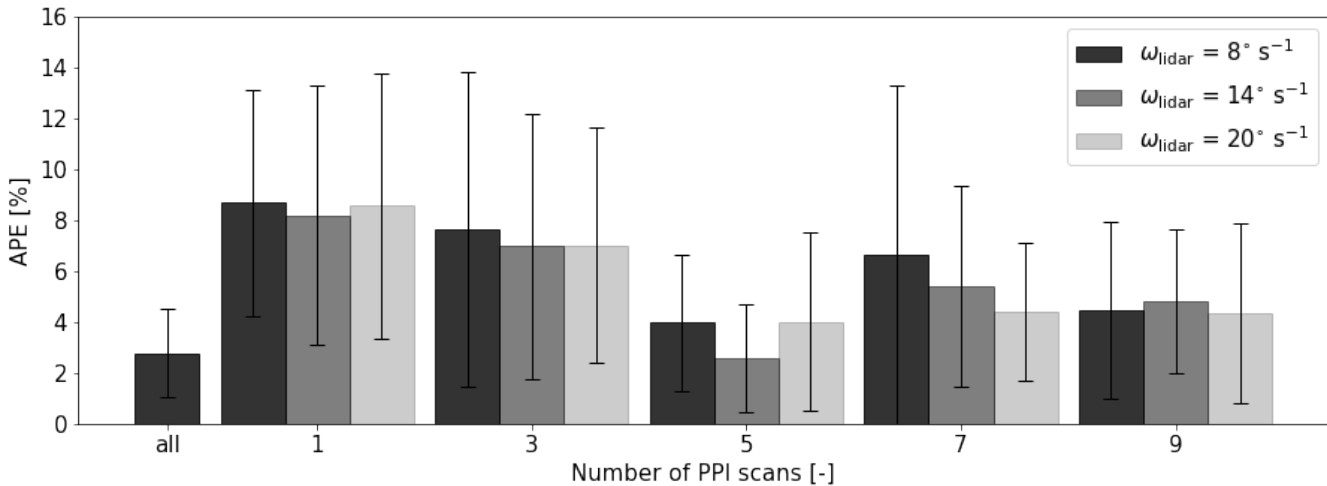

**Figure 2.** Results of the virtual lidar tests. Bars indicate the mean and whiskers the standard deviation of the Absolute Percentage Error (APE) of available power $P_{av}$ over six simulations. The number of PPI scans is indicated on the x-axis. "all" indicates the use of all numerical data, hence the error introduced by the composition method. The opaqueness represents the lidar's angular speed $\omega_{lidar}$.

desirable to measure the largest wake deficit, which is expected to develop around hub height.

Figure 2 shows that 5 PPI scans typically hold the highest accuracy. Using fewer PPI scans results in inaccurate estimations of the wake deficit distribution in the vertical, while using more PPI scans results in long cycles and consequently fewer measurements per observation point. The angular speed $\omega_{lidar}$ seems to have little effect, except for when 7 PPI scans are used. This is attributed to chance, as too few cases are studied for the statistics to converge. Generating more LES results with a wider range of atmospheric conditions and turbine yaw angles was not possible due to computational restrictions. While these results are not statistically significant and it can therefore not be claimed that an 'optimal' scanning strategy is found, this exercise allows for making an informed decision.

It was decided to implement the trajectory showing the lowest error, hence consisting of 5 PPI scans with $\omega_{lidar} = 14°\,s^{-1}$. The elevation angles of these scans were $\phi_{PPI} = (-7.0°, -3.5°, 0.0°, 3.5°, 7.0°)$ and the accumulation time used was 0.1 s. With an opening angle of $70°$, the duration of one PPI scan is 5 s. Changing elevation angles takes 1.3 s and resetting to the start of the cycle takes 3.5 s, adding to 34 s to complete one full cycle. The range gate length was set to 25 m, corresponding to a pulse duration of 100 ns. Range gates were defined between 50 m and 1340 m with 5 m spacing. However, in the processing phase only data up to 820 m were used to avoid the influence of the ground in the PPI scan with the lowest elevation angle.

### 2.3.2 Data processing

Since the performed PPI scans were quite fast with a relatively coarse resolution, all PPI scans with the same elevation angle in a 10-minute window (see Sect. 2.8) were grouped together to get a better estimate of the measurement distribution. Simple filtering based on carrier-to-noise ratio (CNR) and line-of-sight velocity (LOS) was performed, where only realistic

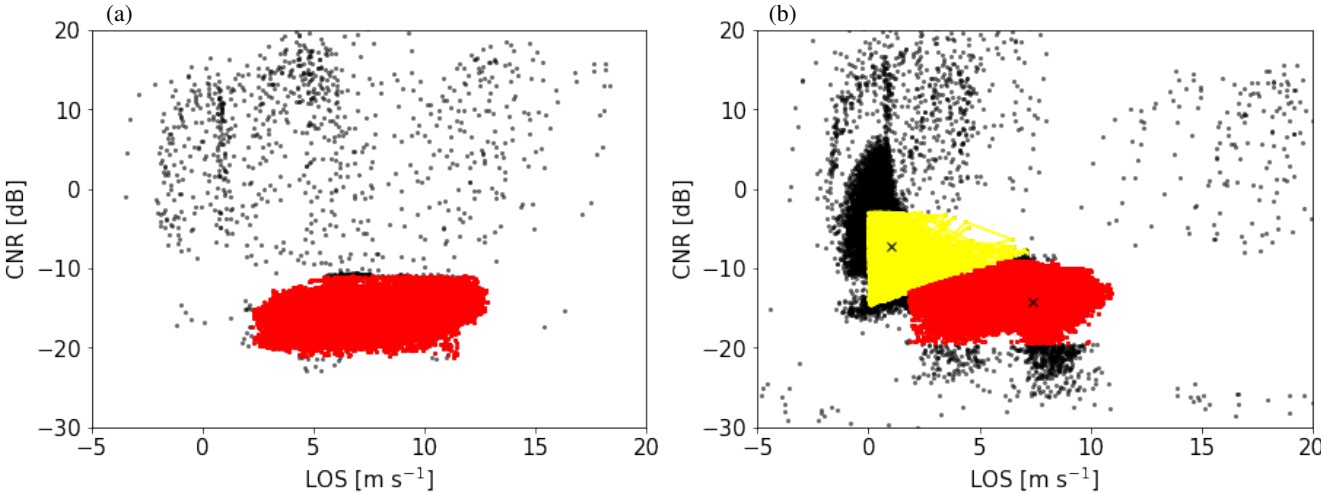

**Figure 3.** Examples of Multiple PPI scan filtering in LOS-CNR diagram in which black markers indicate original data and red markers data kept after filtering. (a) A textbook case with few outliers that indicate hard targets and (b) a more problematic case in which many corrupted measurements. Here yellow markers indicate a second cluster from which all measurements were omitted. Black crosses indicate the two cluster centers.

data with CNR < 0 dB and 0 m s$^{-1}$ < LOS < 20 m s$^{-1}$ were kept. On the remainder, a Gaussian filter was used, retaining only measurements within three standard deviations of the median CNR and LOS (99 % confidence interval). This removed outliers due to hard targets, as illustrated in Fig. 3a.

However, some PPI scans exhibited a LOS-CNR diagram as illustrated in Fig. 3b, containing many measurements with high CNR and low LOS values. To filter out these erroneous measurements, a Mean Shift clustering algorithm (Fukunaga and Hostetler, 1975) was employed, as was for instance used in Wang et al. (2022) as part of data cleaning for power curve tuning. The algorithm identified clusters in the LOS-CNR space and allocated all measurements to any of the clusters based on the Euclidean distance to the cluster center. Clusters were then either considered or eliminated based on whether the location of their center was physically feasible. In the example in Fig. 3b the yellow cluster was omitted, since many points outside the main cluster with high CNR and low LOS values indicate erroneous measurements. Lastly, the Gaussian filter based on the 99 % confidence interval was repeated, as removing one cluster drastically affected the outcome of this filter.

After filtering, all PPI scans were interpolated to a standard grid with a resolution of 1.4° (corresponding to the original resolution) to account for the slightly different azimuth angles between scans as a result of the lidar's inability to measure the exact same location each time. Next, the PPI scans were temporally averaged as long as not more than two data points within a 10-minute window were missing. When more than 25 % of the measurements were filtered out, as is the case with Figure 3b, the averaged PPI scan was removed from the 10-minute window, resulting in fewer than five PPI scans. If fewer than four averaged PPI scans remained after filtering, the case was eliminated.

Lastly, the PPI scans' azimuth and elevation angles were corrected with the nacelle's 10-minute averaged tilt angle and mis-

alignment (see Sect. 2.7). The horizontal wind speed was subsequently computed by correcting the LOS with these azimuth and elevation angles.

## 2.4 Ground-based lidar (VAD)

As shown in Figure 1, the ground-based lidar was situated $1.85D$ upstream of the lidar-equipped turbine for $\delta = 281°$ to measure profiles of wind speed and direction. The ground-based lidar performs continuous velocity-azimuth display (VAD) scans at an elevation angle of $\phi_{\mathrm{VAD}} = 75°$ with an accumulation time of 0.5 s and an angular speed of $30°\,\mathrm{s}^{-1}$. Also for this lidar, the range gate length was set to 25 m, corresponding to a pulse duration of 100 ns. Range gates were defined between 50 m and 840 m with 5 m spacing.

Filtering was done based on the 2D Histogram method introduced by Beck and Kühn (2017), which assumes a normal distribution of LOS and CNR values. The measured data points were binned by their LOS and CNR values and the number of data point in each bin were counted. Bins having a count less than 10 % of the bin with the highest count were omitted.

Next, the azimuth angle ($\theta_{\mathrm{VAD}}$) was corrected by means of a hard target analysis, such that $\theta_{\mathrm{VAD}} = 0°$ faces north. To obtain the wind speed components ($u$, $v$, $w$) and consequently the horizontal wind speed and direction, the measurements of each range gate were fitted with the following sinusoid:

$$\mathrm{LOS} = u \cos(\theta_{\mathrm{VAD}}) \sin(\frac{\pi}{2} - \phi_{\mathrm{VAD}}) + v \sin(\theta_{\mathrm{VAD}}) \sin(\frac{\pi}{2} - \phi_{\mathrm{VAD}}) + w \cos(\frac{\pi}{2} - \phi_{\mathrm{VAD}}) \tag{2}$$

Lastly, only when at least 75 % of the data points remained after filtering and the fitted sinusoid achieved a correlation coefficient of at least 0.8 (determined empirically), the wind speed components of a vertical level were retained.

## 2.5 Met mast

A meteorological (met) mast was positioned $2.7D$ upstream from T1 at $\delta = 350°$ (Fig. 1). This mast was equipped with cup anemometers at 116.3 m (hub height, $U_{\mathrm{h}}$) and 54.2 m (lower tip height, $U_{\mathrm{lt}}$) to measure wind speed and shear. Wind vanes were located at 112.2 m (approximately hub height, $\delta_{\mathrm{h}}$) and 54.5 m (lower tip height, $\delta_{\mathrm{lt}}$). The highest cup anemometer was located on the top of the met mast for undisturbed flow from all directions, whereas the other cup anemometer and wind vanes had orientations of 315° and 135°, respectively. A flow distortion due to the tower structure affecting the measurements occurs for wind directions between approximately 310° and 320°, which is not considered in this study (see Sect. 2.1). The wind directions analyzed here are assumed to be undisturbed. The cup anemometers and vanes had an accuracy of $0.2\,\mathrm{m\,s}^{-1}$ and $1.5°$, respectively. All sensors operated at a sampling frequency of 50 Hz.

## 2.6 Wind turbine operational data

SCADA data were collected at the turbine at a frequency of 50 Hz. These data contain measurements from the nacelle's wind vane $\delta_{\mathrm{S}}$ and cup anemometer $U_{\mathrm{S}}$, as well as power $P$, rotor speed $\omega$ and turbine status, the latter indicating whether the turbine was operating normally. A standard nacelle transfer function was used by the operator to correct wind speed measurements for the influence of the rotor.

## 2.7 GPS

All above-mentioned systems were equipped with a Global Positioning System (GPS) sensor used for time synchronization. Additionally, the nacelle of T1 was equipped with a three-antenna GNSS system to measure orientation, roll and tilt. This system was operated at a sampling frequency of 10 Hz and its measurements have a Root Mean Square Error of less than $0.1°$. This results in a spatial error of less than 1 m at $4D$ downstream.

Orientation measurements, averaged to 10-minute values to smooth out high-frequency vibrations, were used to compute the yaw misalignment $\phi$ of the turbine relative to the wind direction $\delta_h$ measured at the met mast. These measurements were then used to correct the PPI scans' azimuth angles. Likewise, 10-minute averaged nacelle tilt angles were used to correct the PPI scans' elevation angles, but the scans were not corrected for roll as it was expected to only have a small influence on the results.

## 2.8 Selection of data for model evaluation

The measurements were averaged over 10 minutes as is commonly done in the wind energy industry. Case selection was done using the following steps:

1. Within a 10-minute window, no yaw maneuver should take place. A preselection of cases was therefore done purely based on GPS data. A case was considered when the orientation did not change for at least 12 minutes, of which the first two minutes were not considered for analysis because the wake needed time to reach $4D$ downstream. In case the orientation did not change for more than 22 minutes, the first two minutes were omitted and the remainder is split in two 10-minute windows as far apart as possible.

2. The 10-minute averaged $U_h$ needed to be between cut-in and rated wind speed. $\delta_h$ needed to be in the defined sector (Sect. 2.1) and approximately normally distributed. This eliminated situations where there is a clear trend in the wind direction signal.

3. The inflow measured at the met mast should reasonably compare to the measurements at the turbine's nacelle. The met mast measurements were temporally corrected to match the nacelle signal using Taylor's hypothesis of frozen turbulence. Next, the two signals were compared, where the 10-minute averaged wind speed $|U_h - U_S| < 1$ m s$^{-1}$ and direction $|\delta_h - \delta_S| < 5°$.

4. The profiles from the VAD lidar were used to check whether the wind speed profiles were approximately logarithmic, as the effect of low-level jets on the downstream wake characteristics was currently not captured by the wake models and considered out of the scope of this study.

5. If all checks were passed, all completed cycles within the defined 10-minute window were averaged as described in Sect. 2.3. After averaging, the PPI scans were interpolated to a vertical plane at $4D$ downstream of the turbine. The wake deficit ($U_{def}$) was calculated by subtracting the wake measurements with the inflow profile obtained from the met mast measurements, and normalized by dividing by the hub height wind speed $U_h$.

6. Lastly, the 10-minute averaged cases were evaluated by the Multiple 1D Gaussian method (see Sect. 3.1). Since the opening angle of the PPI scans is 70°, it can be expected that wakes from other turbines are also visible in the measurements. To prevent using an incorrect wake, the scans are sliced around the expected location of the considered wake. Boundaries of these slices are determined by the maximum wind speeds between the scan's center, corrected for yaw misalignment, and 150 m left and right of this center. Furthermore, the correlation coefficient (R) of the Gaussian fit with the wake deficit observations needed to be higher than 0.85 (empirically determined) to be considered, removing cases that do not fulfill the model assumptions of a Gaussian wake deficit.

This selection procedure resulted in 382 10-minute averaged cases to be used for analysis. Figure 4 displays the distribution of measured yaw angles during the campaign. Most measurement were done without yaw misalignment, since during a part of the campaign the implemented controller had issues and turbine control reverted back to standard operation. The difference between the number of positive and negative yaw angles is due to a more dominant wind direction in the sector containing positive yaw angles.

The solid vertical lines indicate the median yaw angles per target angle. For greedy control, the median shows a small bias of $\phi = -0.94°$, suggesting a calibration error of the nacelle's wind vane. For a target angle $\phi_t = +15°$, the median achieved $\phi = +11.14°$, whereas for $\phi_t = -15°$, $\phi = -13.19°$ is achieved. These angles are smaller than the targeted angles, which is due to the wind vane error under yaw misalignment (Kragh and Fleming, 2012; Simley et al., 2021a). Figure 5 displays an overview of the inflow conditions measured during these 382 cases. The shear $\alpha$ with a mean of 0.3 is slightly larger than

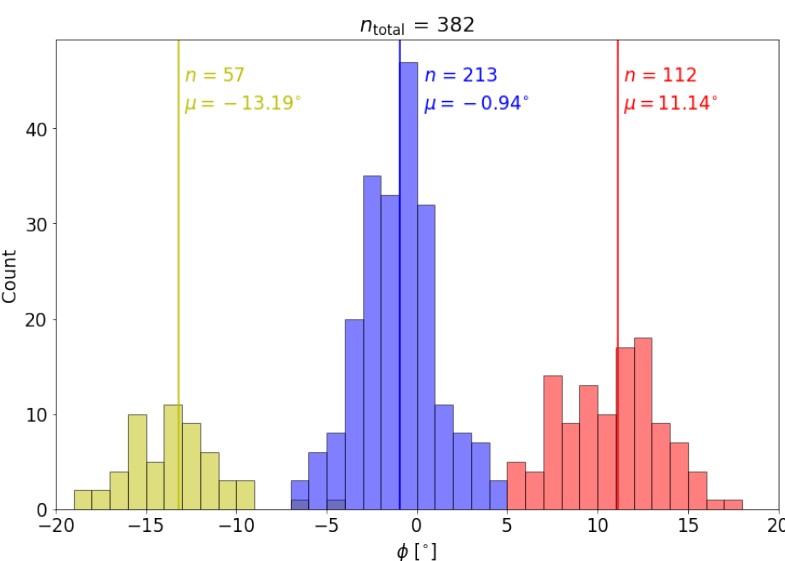

**Figure 4.** Data availability of the 10-minute averaged cases as a function of achieved yaw angle ($\phi$). Colors indicate the targeted $\phi_t = -15°$ (yellow), $\phi_t = 0°$ (blue) and $\phi_t = +15°$ (red). Solid vertical lines and accompanying text mark the median of the achieved yaw angles.

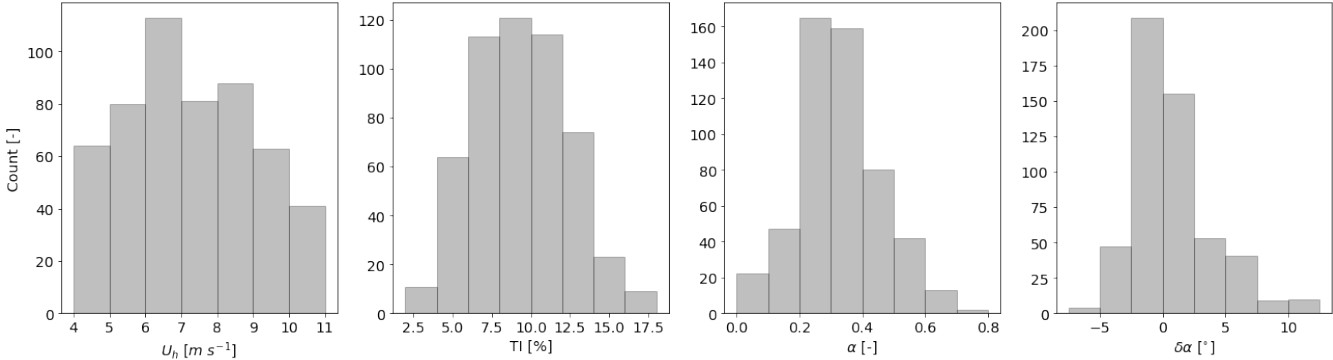

**Figure 5.** Distribution of 10-minute averaged inflow variables measured at the met mast for all 382 cases: hub height wind speed ($U_{\mathrm{h}}$), turbulence intensity (TI), shear ($\alpha$) and veer ($\delta\alpha$).

expected and the veer $\delta\alpha$ is smaller than expected, showing a high occurrence of negative values. Regardless, all variables show a range of values that are physically reasonable.

## 3 Methods

This section introduces the modeling aspects of this study. First, Sect. 3.1 summarizes the Multiple 1D Gaussian method used
290 to obtain quantifiable wake characteristics. Sect. 3.2 discusses what information is used as a reference and Sect. 3.3 describes the splitting of the data set in training and testing subsets. Then, Sect. 3.4 introduces the data-driven model and Sect. 3.5 briefly introduces the analytical model used in this study.

### 3.1 Multiple 1D Gaussian method

The Multiple 1D Gaussian method (Sengers et al., 2020) is utilized to obtain quantifiable wake characteristics, listed in Table 1.
295 This method fits a 1D Gaussian through the wake deficit data normalized by the wind speed at hub height ($U_{\mathrm{def}}/U_{\mathrm{h}}$) in the horizontal plane for every height level, in the current study obtained from five consecutive PPI scans. This results in a set of local wake deficits (amplitude), center positions (location) and widths (standard deviation) for each height. By fitting another 1D Gaussian through the set of local deficits in the vertical, the vertical deficit profile can be determined. The position of the maximum deficit in this profile is then considered as the vertical position of the wake center. The horizontal position of the
300 wake center is determined by interpolating the set of local center positions to this height. A second-order polynomial is fit through the set of local wake center positions to find the wake curl and tilt. The same method is applied for the wake widths to find their profile as function of height.

The reverse of this method, hereafter called composition method, can be used to obtain a vertical cross-section of the wake from a set of wake characteristics. For more details on the Multiple 1D Gaussian method and the composition method, the
305 reader is referred to Sengers et al. (2020, 2022).

**Table 1.** Dimensionless variables describing the wake characteristics obtained with the Multiple 1D Gaussian method. Reused from Sengers et al. (2022) with permission.

| Scalar Parameter | Symbol |
| --- | --- |
| Amplitude normalized wake deficit | $A_z$ |
| Lateral wake center displacement | $\mu_y$ |
| Vertical wake center displacement | $\mu_z$ |
| Width wake center height | $\sigma_y$ |
| Vertical extend | $\sigma_z$ |
| Curl | curl |
| Tilt | tilt |
| Quadratic wake width parameter | $s_a$ |
| Linear wake width parameter | $s_b$ |

## 3.2 Reference power

The wind speed measured by the nacelle lidar is used to obtain the available power at $4D$ downstream. Since the spatial resolution is relatively coarse and the two outermost PPI scans target the tip heights, the 10-minute averaged wind speeds are interpolated using a cubic spline function to a resolution of $\Delta = 5$ m. This inherently fills gaps when data are not available. The spatially interpolated data are consequently used to determine a rotor equivalent wind speed ($U_{\mathrm{eq}}$) and an available power $P_{\mathrm{av,ref}}$ used as a reference in the remainder of this study.

## 3.3 Training and testing data

The data set is split into a training part (80 % of total size) and a testing part (remaining 20 %). This has been done in a stratified random manner, meaning the data set was first split up in three subsets according to their target yaw angle $\phi_{\mathrm{t}} = (-15°, \ 0°, \ 15°)$, after which from each subset 20 % was randomly selected to be testing data. This way, it is ensured that both testing and training data contain cases with a yaw misalignment.

To not base the results on only a single testing data set, this randomly splitting of the data set (resampling) has been repeated 96 times (hereafter: resamples). The choice of 96 resamples is pragmatic, as it was convenient for parallel computing (multiple of 24 nodes per core). Error statistics appear to be normally distributed, which was not yet the case with 24, 48 or 72 resamples. Although more resamples are desirable (e.g., bootstrapping is typically done over several thousands), this was not possible due to computational limitations as the training of the models can be quite expensive as discussed in Sect. 3.4.4 and 3.5.

### 3.4 Data-driven wAke steeRing surrogaTe model (DART)

This section introduces DART, starting with a summary from previous work in Sect. 3.4.1 and changed made to the model since this work in Sect. 3.4.2. This is followed by information on the input variables (Sect. 3.4.3). Lastly, the feature selection of the three version of the model considering in this study is discussed in Sect. 3.4.4.

### 3.4.1 Model description

DART was introduced in an LES study by Sengers et al. (2022). It estimates wake characteristics (Table 1) obtained from the Multiple 1D Gaussian method (Sect. 3.1) with a linear regression model from standard input parameters (e.g., yaw misalignment, shear, thrust coefficient). These wake characteristics ($\mathbf{Y}$) are estimated from input parameters ($\mathbf{X}$) using a simple linear model:

$$\underset{(n)}{\mathbf{Y}} = \underset{(n \times p)}{\mathbf{X}} \times \underset{(p)}{\mathbf{B}}. \tag{3}$$

in which $\mathbf{B}$ are the model coefficients. The matrix dimensions are indicated by the sample size $n$ and the number of input parameters $p$, containing the input variables, their second-order and interaction terms, as well as intercepts. The model coefficients are fitted with the Lasso method (Tibshirani, 1996), using the following cost function:

$$\underset{B}{\mathrm{argmin}} \sum_n (y_n - \sum_p x_{np} \mathbf{B}_p)^2 + \lambda \sum_p |\mathbf{B}_p|. \tag{4}$$

This method remains close to ordinary least squares, but adds a regularization parameter $\lambda$ to its cost function, effectively penalizing adding more parameters. This ensures shrinkage of the input parameters and eliminates the issue of multicollinearity as only one of the highly correlated input parameters is chosen. The notations presented here deviate slightly from those in Sengers et al. (2022), as in the current study only one distance downstream is considered, simplifying the equations.

To include nonlinear relations between input parameters and wake characteristics, the original variables can be transformed with e.g., a square-root or exponential transformation. In the training stage (Sect. 3.4.4), it is determined what set of input variables and transformations yields the most accurate results.

Lastly, the estimated wake characteristics are used in a composition method (Sect. 3.1) to generate a vertical cross-section of the wake deficit and wind field. For a more detailed description of DART, the reader is referred to Sengers et al. (2022).

### 3.4.2 Modifications to the model

A few changes have been made to DART since its first description in Sengers et al. (2022). Most notably, the feature selection procedure has been changed. Before estimating the wake characteristics with a linear model, inflow variables (e.g., $\phi$, $\alpha$, $\omega$) undergo transformations. In addition to the non-transformed variable, the square root, exponent, natural logarithm and reciprocal transformation are considered for all input variables, resulting in five options for each variable and many possible sets of input parameters (e.g., $\phi$, $\alpha^{-1}$, $\ln(\omega)$). In Sengers et al. (2022), all these possibilities were tested, the available power of a virtual turbine was estimated and compared to the original data, and the set of input parameters that had the smallest error was chosen.

This selection procedure was not only very computationally expensive, but also does not necessarily give the most accurate solution for all wake characteristics. Hypothetically, the wake center position could be best explained by the non-transformed yaw angle, whereas the wake curl could be best explained by the exponent of the yaw angle. In the current work, the determination of the best set of transformations is tested for each wake steering variable individually. The best transformation is then chosen as the one that has the smallest mean absolute error on the training data. This not only allows for more accurate estimates, but also speeds up the training process.

Secondly, square root and natural logarithm transformations do not allow for negative input values. A sign function is used to include these values rather than omitting them, as was done in the previous work.

Lastly, in the testing phase extrapolation is prevented by using the maximum (or minimum) value found in the training data when an input variables exceeds this range. Although this does not allow DART to give accurate estimations in new situations, it eliminates erroneous estimates due to extrapolation.

### 3.4.3 Input variables

As argued in Sengers et al. (2022), highly correlated input variables are interchangeable as they provide similar information. However, as long as they are not perfectly correlated, including all variables can lead to a higher accuracy as some new information is added. Due to the use of the Lasso regression method, multicollinearity is not an issue.

Because of DART's flexibility, training with different sets of input variables is possible, allowing for an analysis of the model's accuracy as a function of chosen input variables. An overview of the available input variables is displayed in a correlation matrix (using the Pearson correlation coefficient) in Fig. 6. Other variables such as the wind direction variability and TI at

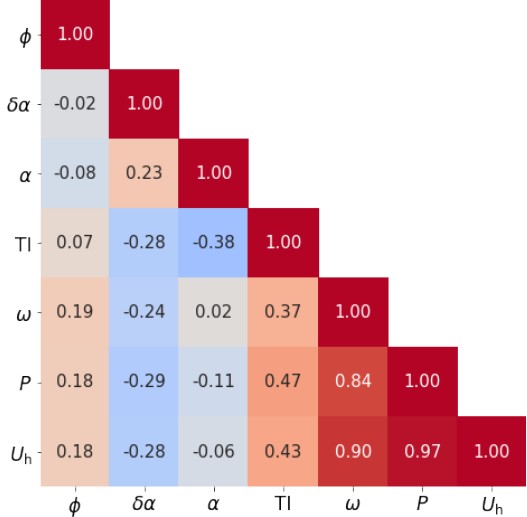

**Figure 6.** Correlation matrix of available input variables. In addition to the inflow variables shown in Fig. 5, yaw misalignment $\phi$, rotor speed $\omega$ and power $P$ are considered input variables.

lower tip height could have been included, but were omitted for brevity. As opposed to what was seen in LES in Sengers et al. (2022), the inflow variables $\delta\alpha$, $\alpha$ and TI are weakly correlated in this field experiment. Secondly, $\omega$ and $P$ are highly correlated with $U_{\mathrm{h}}$.

### 3.4.4 Feature selection

In this study, multiple versions of the DART model were considered, each having a different set of input variables. Adding
more input variables might increase the accuracy, but will increase the training time of the model significantly. In Sengers et al. (2022) it was hypothesized that DART can achieve reasonable accuracy as long as each of the following clusters was represented: yaw ($\phi$), atmospheric inflow ($\delta\alpha$, $\alpha$, TI) and turbine ($\omega$, $P$, $U_{\mathrm{h}}$). Due to its high correlation with the turbine variables, $U_{\mathrm{h}}$ is here considered a turbine variable rather than an inflow variable. Following this logic, the first version of DART uses three input variables.


**DART-3**

To determine the most accurate solution using only three variables, all possible sets of input variables and their respective transformations (Sect. 3.4.2) are tested during the training stage and their accuracy to reproduce the training data set is investigated. By means of an example, Fig 7a displays the error distribution of one resample. The error metric used here is the Percentage
Error (PE) of $P_{\mathrm{av}}$ (calculated analogous to Eq. (1), but without absolute values) at $4D$ downstream. From these values, a Mean Percentage Error (MPE) and Mean Absolute Percentage Error (MAPE) can be computed, as indicated in the top right of the figure.

Repeating this for all 96 resamples, one can obtain a histogram of MPEs and MAPEs as displayed in Fig. 7b and c. Finally, the mean over 96 MPEs ($\overline{\mathrm{MPE}}$) and MAPEs ($\overline{\mathrm{MAPE}}$) can be calculated, see top right of the figures. The $\overline{\mathrm{MPE}} = -2.19$ %
illustrates a negative systematic bias, meaning DART underestimates $P_{\mathrm{av}}$.

To determine the most accurate set of input variables, $\overline{\mathrm{MAPE}}$ is considered. The results for all considered combinations of input variables are displayed in Table 2, showing that the set ($\phi$, $\alpha$, $P$) provides the most accurate result (lowest $\overline{\mathrm{MAPE}}$) and is therefore used in the remainder of the study, denoted as DART-3. The training time for each set of input variables with DART-3 is in the order of 10 minutes, hence the total computation time to determine the best set of input variables is approximately 1.5 hours.


**DART-4**

Because the training of DART-3 is fast, an additional variable can be included to improve the accuracy of the model. The first three variables are chosen similarly to DART-3 (one from each cluster), while the fourth variable can be any input variable not yet selected. Repeating the analysis of computing a MAPE for each resample and consequently a $\overline{\mathrm{MAPE}}$ for each set of input
variables, generating a table corresponding to Table 2 (not shown here for brevity), reveals that ($\phi$, $\alpha$, $P$, $U_{\mathrm{h}}$) is the most accurate combination with $\overline{\mathrm{MAPE}} = 12.69$ %, hereafter called DART-4. Its training time for each combination is approximately one hour, hence with 18 possible sets of input variables the total computation time needed for training is 18 hours.

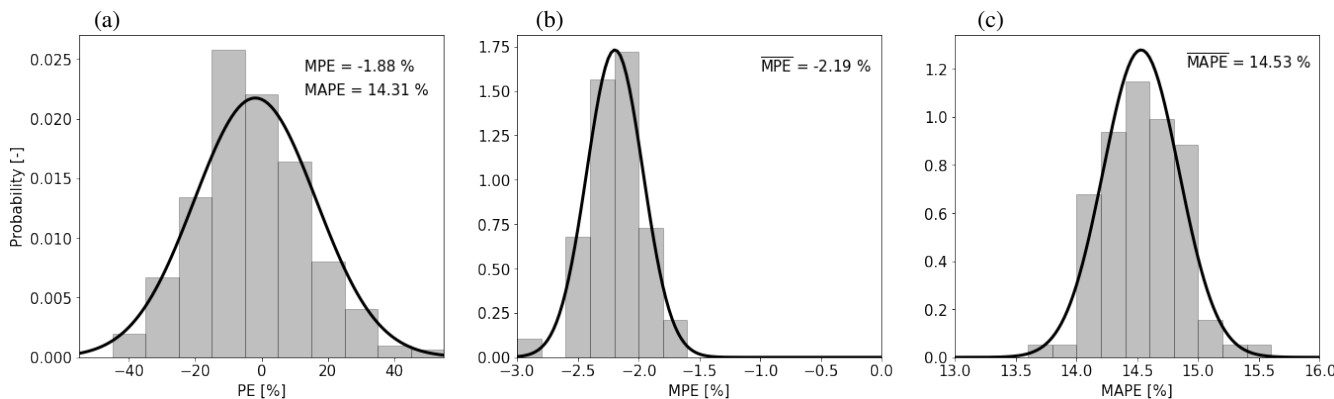

**Figure 7.** Performance of DART-3 on the training data. The set of input variables is $(\phi, \alpha, P)$. (a) Histogram of PE of $P_{\mathrm{av}}$ for one resample. Fitted normal distributions are indicated with solid lines and MPE and MAPE are given in the top right. Histograms of MPEs (b) and MAPEs (c) over all 96 resamples.

**Table 2.** Overview of all possible combinations of input variables in DART-3 and their respective $\overline{\mathrm{MAPE}}$ values. Boldface indicates the combination resulting in the lowest error.

| Variable 1 | Variable 2 | Variable 3 | $\overline{\mathrm{MAPE}}$ [%] |
|---|---|---|---|
| $\phi$ | $\delta\alpha$ | $\omega$ | 18.24 |
| $\phi$ | $\delta\alpha$ | $P$ | 16.70 |
| $\phi$ | $\delta\alpha$ | $U_{\mathrm{h}}$ | 17.17 |
| $\phi$ | $\alpha$ | $\omega$ | 15.11 |
| $\boldsymbol{\phi}$ | $\boldsymbol{\alpha}$ | $\boldsymbol{P}$ | **14.53** |
| $\phi$ | $\alpha$ | $U_{\mathrm{h}}$ | 15.08 |
| $\phi$ | TI | $\omega$ | 15.66 |
| $\phi$ | TI | $P$ | 14.77 |
| $\phi$ | TI | $U_{\mathrm{h}}$ | 14.81 |

### DART-7

Lastly, all available variables are used as input in DART-7, demonstrating the maximum achievable accuracy of the data-driven model during this experiment. DART-7's accuracy on the training data was indeed the highest with $\overline{\mathrm{MAPE}} = 10.31$ %. The computation time needed to train DART-7 is approximately one month if not parallelized.

### 3.5   Analytical wake model

The state-of-the-art GCH model (King et al., 2021) as available in version 3.0rc4 of the FLORIS framework (NREL, 2022)
acts as a reference model in this study. The GCH model incorporates the spanwise and vertical velocity components (Martínez-Tossas et al., 2019) due to the present vortices to the Gaussian wake model (Bastankhah and Porté-Agel, 2014, 2016; Niayifar

and Porté-Agel, 2016).

Since presently only the wake at a distance of $4D$ behind the upstream turbine is studied, the GCH model could have benefited from including a near-wake model (e.g., Blondel and Cathelain, 2020), but this coupling was not available in this version of FLORIS. The $C_T$ curve of the *eno126* turbine is obtained from the Bladed model for which the aerodynamic properties of the turbine were provided by the operator, and is used in these calculations.

Inflow information is taken from the 10-minute averaged met mast data. The model tuning parameters ($\alpha_{GCH}$, $\beta_{GCH}$ for the far-wake onset, and $k_{a,GCH}$, $k_{b,GCH}$ for the wake growth rate) are determined by minimizing the MAPE of available power over the training data, analogous to the training of DART described in Sect. 3.4.4. The tuning takes about 3 hours and the model has an error of $\overline{\text{MAPE}} = 18.13$ %.

## 4 Results

This section presents the results of this study. Section 4.1 describes the characteristics of the wake observed in the field, after which in Sect. 4.2 the performance of the wake models in reproducing these wake characteristics is discussed.

### 4.1 Observed wake characteristics

In Sect. 4.1.1 an assessment of the characteristics of the observed wake listed in Table 1 is performed, which is deemed a necessary first step before investigating the accuracy of wake models. The observed wake characteristics are linked to the inflow variables to examine whether the measurements are physically feasible. In Sect. 4.1.2, two wake characteristics that are deemed important for wake steering are further investigated.

#### 4.1.1 Correlation with inflow variables

The Multiple 1D Gaussian method (Sect. 3.1) is used to describe the wake in quantifiable characteristics. Figure 8 displays how the nine wake characteristics correlate with the input variables.

The wake center deficit normalized with the hub height wind speed, denoted $A_z$, is highly correlated with shear $\alpha$ and turbulence intensity TI and shows a moderate correlation with veer $\delta\alpha$, corresponding to the correlations found in previous studies (e.g., Bastankhah and Porté-Agel, 2016; Schottler et al., 2017). $A_z$ has a weak correlation with the hub height wind speed $U_h$ as it was already used to normalize the deficit.

The lateral wake center displacement $\mu_y$ has a relatively high correlation with the yaw misalignment $\phi$, confirming that the wake is deflected when the turbine is operated with a yaw misalignment. Moderate correlations with $\alpha$ and $\delta\alpha$ are found, corresponding to previous findings (e.g., Fleming et al., 2015; Sengers et al., 2022) that found that wake deflection is affected by atmospheric conditions. The vertical wake center displacement $\mu_z$, a relatively unexplored wake characteristic, appears to be positively correlated with $\alpha$. It is hypothesized that this is due to a larger wind speed gradient at lower tip height, increasing the mixing compared to that at upper tip height, effectively moving the wake center upwards. Further analysis (results not shown here) suggests that the vertical wake center displacement, and with that its correlation with input variables, is independent of

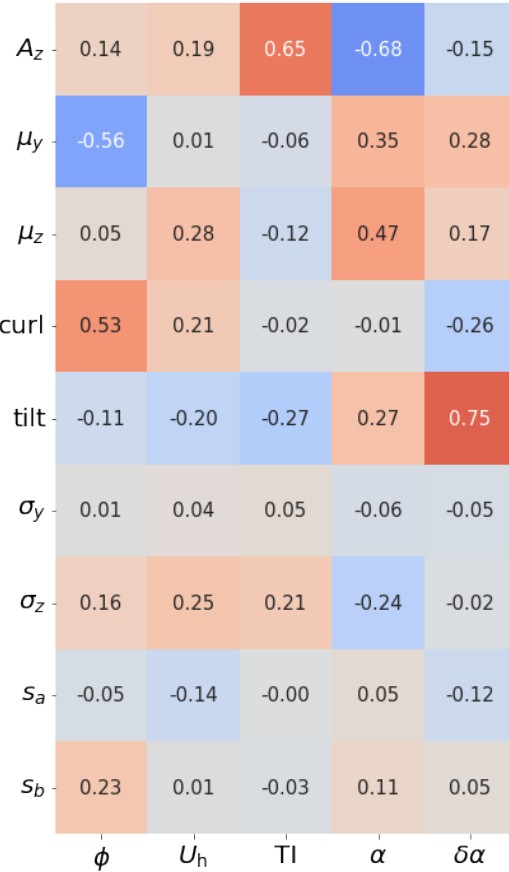

**Figure 8.** Correlation matrix of the input variables and wake characteristics. $A_z$ is the amplitude of the wake deficit normalized by $U_h$, $\mu_y$ and $\mu_z$ the lateral and vertical wake center displacement, the wake curl and tilt, $\sigma_y$ and $\sigma_z$ the width and height of the wake and $s_a$ and $s_b$ the quadratic and linear wake width parameter, respectively.

wind direction. This excludes the influence of topography on these results.

The curl only correlates with $\phi$, whereas the wake tilt is highly correlated with $\delta\alpha$, corresponding to Abkar et al. (2018). Lastly,
variables related to wake size ($\sigma_y$, $\sigma_z$, $s_a$, $s_b$) have very weak correlations with the input variables, which could be due to the spatial resolution of the lidar PPI scans.

### 4.1.2 Lateral wake center displacement and wake curl

Two wake characteristics, $\mu_y$ and curl, are investigated as function of $\phi$ as these are deemed important for wake steering. Figure
9a demonstrates that $|\mu_y|$ typically increases with $|\phi|$, hence the wake deflection is larger for larger yaw misalignment angles, although there is a lot of scatter in the field measurements as also indicated by the correlation coefficient R. Three clusters can

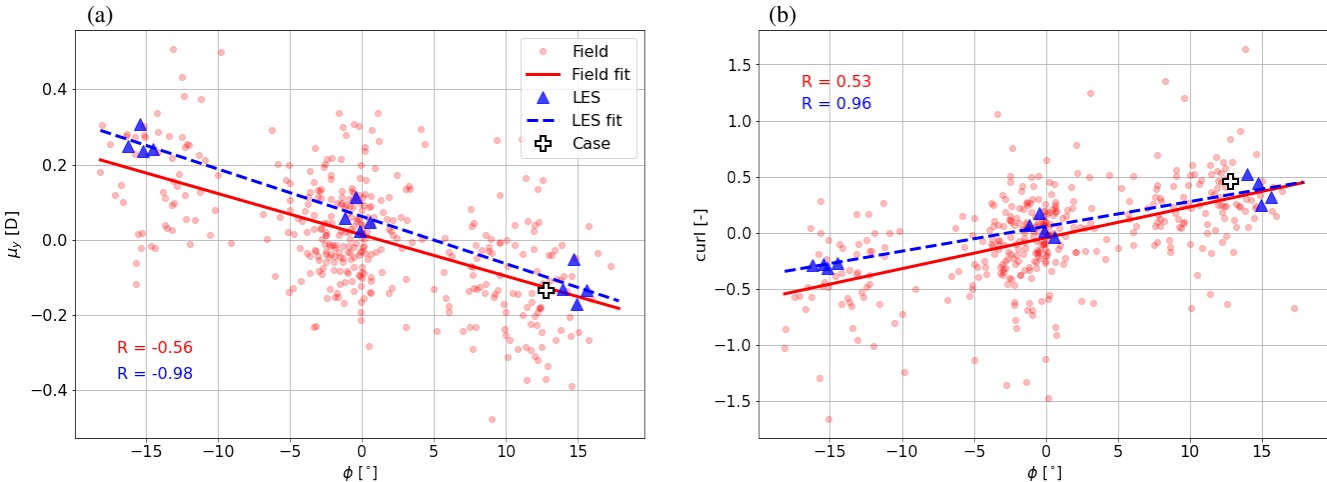

**Figure 9.** Scatter plot of (a) $\mu_y$ and (b) curl as a function of yaw angle $\phi$. Red markers indicate field measurements and blue triangles indicate LES data. Fitted linear functions are indicated with lines. The quality of these fits is indicated by the correlation coefficient R, corresponding to Fig. 8. White plus signs indicate the case studied in Fig. 10.

be identified, corresponding to the distribution of yaw angles shown in Fig. 4.

To check whether $\mu_y$'s order of magnitude is reasonable, field measurements are compared with LES results. Different than in Sect. 2.3, the turbine simulated here represents turbine T1 in the field, for which the aerodynamic properties were provided

by the operator in the Bladed model and translated into FAST. Because of computational restrictions, only three yaw settings $(-15°, 0°, 15°)$ with each four inflow conditions were simulated, which will represent only a small part of the full range of conditions observed in the field. The simulations have $U_h \approx 8 \text{ m s}^{-1}$, and the inflow variables are $0.11 < \alpha < 0.26$; $1.1° < \delta\alpha < 2.6°$ and $6.0\% < \text{TI} < 8.4\%$. The LES results show an initial deflection for $\phi = 0°$ (Gebraad et al., 2016), which is not clearly observed in the field. Otherwise, the observed magnitude of deflection is comparable between LES and the field.

Figure 9b displays curl as function of $\phi$. Similar to $\mu_y$, the field measurements have a larger spread than the LES results, expressed by the lower quality of the linear fit (correlation coefficient R). However, the fitted lines are similar, indicating that the wake curl does indeed occur in field, something that until now had not conclusively been shown in literature.

One case is selected (indicated with a white plus sign in Fig. 9) to illustrate what a wake with curl $\approx 0.5$ looks like. Figure 10a presents the observed deficit measurements ($U_{\text{def}}$) normalized by $U_h$, in which the wake's curl is indicated by the black dashed

line. The curl is indeed relatively small and could be missed when operating the long-range lidar with a different scanning strategy. Figure 10b represents a reconstructed wake using the composition method (Sect. 3.1), which clearly shows that the wake center has moved to the left and up.

Even though the curling observed in this study is relatively small, Figure 9b does confirm that the wake curls as expected from numerical and wind tunnel experiments. Field experiments are often restricted to yaw misalignments smaller than $20°$, whereas

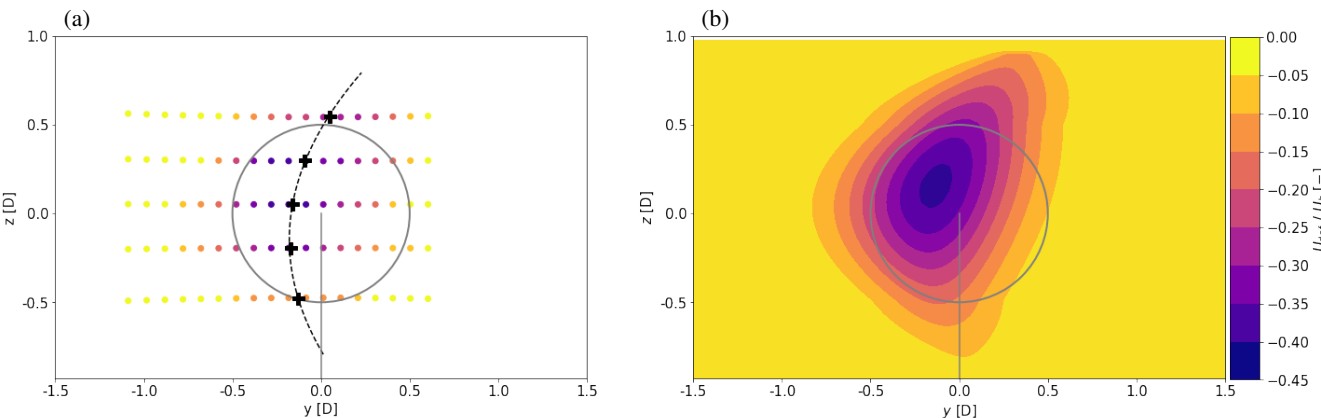

**Figure 10.** Exemplary case illustrating the wake curl generated by a misaligned turbine ($\phi = 12.8°$, $U_\mathrm{h} = 9.4\,\mathrm{m\,s^{-1}}$, $\alpha = 0.33$, $\delta\alpha = -0.8°$, TI $= 10.4\%$). (a) The wake deficit of the ten-minute averaged lidar data of 17 consecutive PPI scans (colors) and local wake center positions (black plus signs) with corresponding fitted polynomial (black dashed line) indicating the wake curl. (b) The reconstructed wake by utilizing the composition method. The colorbar applies to both figures.

numerical and wind tunnel studies allow for larger misalignments. As suggested in Brugger et al. (2020), this is the reason for the lack of observations of fully curled (or kidney-shaped) wakes in the field.

## 4.2 Performance of wake models

This section presents the performance of the DART and GCH wake models in reproducing the wake characteristics observed in the field. Section 4.2.1 presents how well the models can reproduce the available power measured by the lidar. Following

this general result, Sect. 4.2.2 zooms in on how well the model perform under different conditions and Sect. 4.2.3 displays how well the models can reproduce a selection of wake characteristics. Section 4.2.4 discusses how sensitive the models are to the amount of training data. Lastly, Sect. 4.2.5 evaluates how well DART performs when only using SCADA data as input.

### 4.2.1 Comparison of DART and GCH

This section discusses the performance of DART and GCH in a comparison with the wake observed in the field. The models

were trained (DART, Sect. 3.4.4) or tuned (GCH, Sect. 3.5) on 80 % and are now tested on the remaining 20 % of the data. Fig. 11a displays the model accuracy on one resample, using the Percentage Error (PE) of available power ($P_\mathrm{av}$) as a performance metric, analogous to Fig. 7. All models seem to have negative bias (MPE < 0 %), indicating that the rotor equivalent wind speed $U_\mathrm{eq}$ is overestimated. DART's bias reduces with increasing number of input variables, as is evident from the smaller error of DART-7 compared to DART-3 and DART-4. DART-7's bias is comparable to that of GCH. GCH's spread is however

larger than DART's, resulting in a larger MAPE despite having a lower MPE.

When repeating this for all 96 resamples, a distribution of MPE and MAPE values can be found (Fig. 11b-c). Also here DART shows a small negative bias ($\overline{\mathrm{MPE}} < 0$ %), hence underestimating $U_\mathrm{eq}$. GCH has a small positive bias, therefore overestimat-

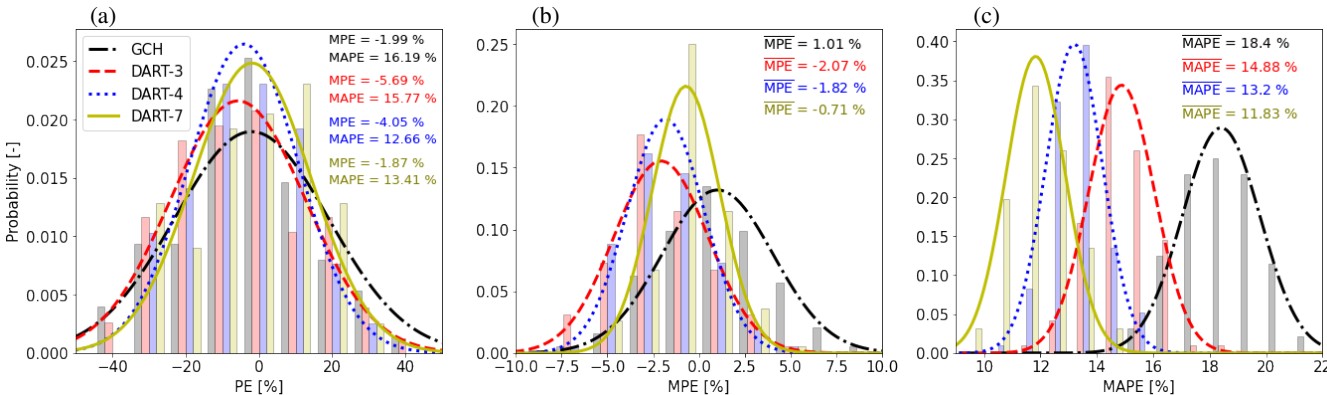

**Figure 11.** Like Fig. 7, but for the testing data for GCH (black/grey), DART-3 (red), DART-4 (blue) and DART-7 (yellow). (a) Histogram for one resample, (b) for MPEs and (c) for MAPEs of all 96 resamples. Fitted normal distributions are indicated with lines and statistics are given in the top right.

ing $U_{\text{eq}}$, and a much wider distribution.

The distribution of MAPE values indicates that with just three input variables, DART-3 is able to outperform GCH, showing
a reduction of $\overline{\text{MAPE}}$ of 19 %. Moreover, both its $\overline{\text{MPE}}$ and $\overline{\text{MAPE}}$ are very similar to those of the training data (Fig. 7), indicating that the model is able to generalize well to unseen or independent data. Adding more variables further improves DART's accuracy, reducing $\overline{\text{MAPE}}$ with 28 % and 36 % for DART-4 and DART-7 compared to GCH. Moreover, the fitted normal distributions of $\overline{\text{MAPE}}$ for GCH and DART-4 or DART-7 are hardly overlapping, indicating that DART significantly outperforms GCH when trained with at least four variables.

These results show the potential of a data-driven model: more of the variability of wakes observed in the field can be explained by using only four input parameters in a data-driven model than with an industry-standard analytical model.

### 4.2.2 Model accuracy under different conditions

To gain a better understanding of these results, it is investigated under what conditions the models' performances differ con-
siderably. DART-3 is here omitted for brevity. First, the models' errors are investigated in relation to yaw angle $\phi$. Figure 12a displays a histogram of data availability per $\phi$ bin of $5°$ over all 96 resamples, while Fig. 12b and c show the MPE and MAPE of $P_{\text{av}}$ per bin. The GCH model has MPE > 0 % for $\phi < -7.5°$ and MPE < 0 % for $2.5° < \phi < 7.5°$. This is likely due to low data availability. DART demonstrates a more uniform trend over all yaw angles. When looking at MAPE (Fig. 12c), it can be seen that especially under yawed conditions (both positive and negative) DART seems to outperform GCH. It is hypothesized
that this is due to a more accurate estimation of the wake center position in DART.

Figures 12d-f display a similar analysis as function of shear $\alpha$. GCH shows an almost linear trend as function of $\alpha$, with MPE < 0 % for small $\alpha$ and MPE > 0 % for large $\alpha$. This indicates that the modeled wake recovery is too slow under low

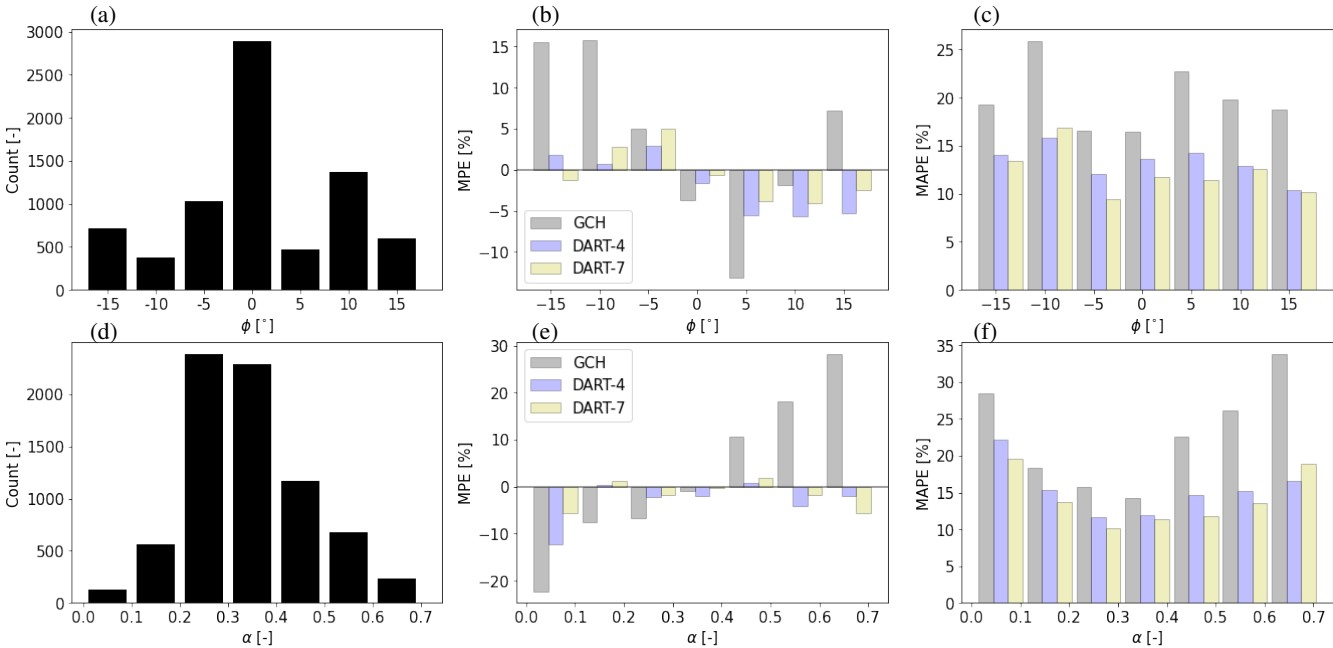

**Figure 12.** Performance of GCH (grey), DART-4 (blue) and DART-7 (yellow) as function of $\phi$ (a-c) and $\alpha$ (d-f). Histogram of data availability per bin (a,d) and corresponding MPE (b,e) and MAPE (c,f) per bin.

shear and too fast under high shear inflow, which could be due to the turbulence model not explicitly including $\alpha$ as an input parameter. In contrast, DART explicitly uses $\alpha$ to estimate wake characteristics. It therefore produces more uniform results and outperforms GCH especially when $\alpha > 0.4$ (see Fig. 12c). Over the whole range of $\phi$ and $\alpha$, DART-7 is marginally more accurate than DART-4.

### 4.2.3 Estimating wake characteristics

Finally, the accuracy of GCH and DART in estimating the wake characteristics $A_z$, $\mu_y$ and curl is investigated. The left column of Figure 13 displays the observed $A_z$ as a function of the model estimated $\hat{A}_z$. As clearly indicated by the fitted line, the GCH model overestimates small deficits and underestimates large deficits. This could be resolved by giving more weight to outliers when tuning the parameters, although that could lead to the undesirable decrease of accuracy in frequently occurring conditions. The fitted line to the DART-4 results appears to be closer to the unity line, while for DART-7 an even better agreement is found. Additionally, the Mean Absolute Error (MAE) and Pearson correlation coefficient (R) displayed in the top left indicate a more accurate modeling of $A_z$ using DART.

The center column displays a similar analysis for $\mu_y$. The fitted lines imply a higher accuracy for GCH than for DART, while the statistical metrics suggest the opposite. GCH's estimates for $\mu_y$ seem to be clustered, which is not true for DART or the

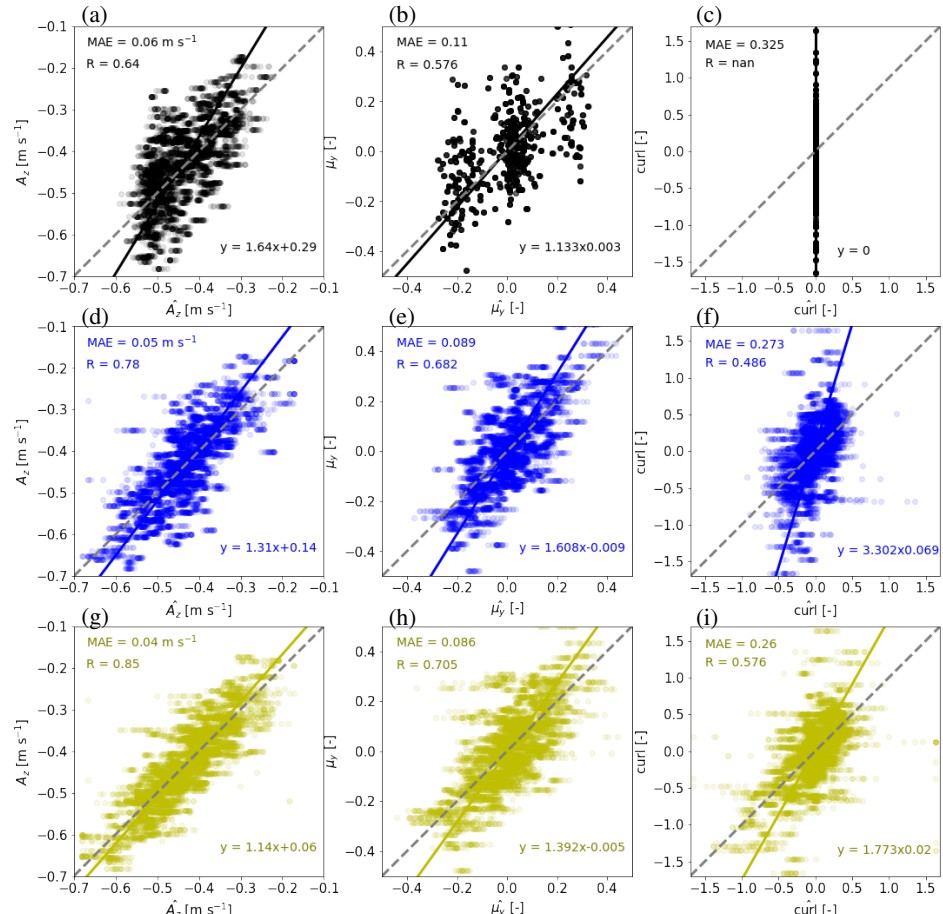

**Figure 13.** Accuracy of GCH (a-c), DART-4 (d-f) and DART-7 (g-i) in estimating wake characteristics $A_z$ (a,d,f), $\mu_y$ (b,e,h) and curl (c,f,i). The models' estimates are given on the x-axis, the observations on the y-axis. Solid lines indicate linear Orthogonal Distance Regression fits and dashed lines the identity lines.

measurements. As noted in Sengers et al. (2022), the effect of inflow conditions (e.g., $\alpha$) on the wake deflection is not well described in GCH. Consequently, the wake deflection is only a function of yaw angle and the observed clusters can directly be related to the distribution of yaw misalignments angles shown in Fig. 4. Additionally, while in Fig. 13e and Fig. 13h transparent markers can be observed, the markers in Fig. 13b appear to be opaque. Here, many transparent markers overlay each other, indicating that GCH estimates the same wake center location in all resamples and that these estimates are not affected by the model's tuning parameters.

Lastly, the right column displays the results for curl. GCH does not model any curl, whereas DART is able to capture some of the observed variability. DART-7 performs better than DART-4 as more variables that are (weakly) correlated with curl (see

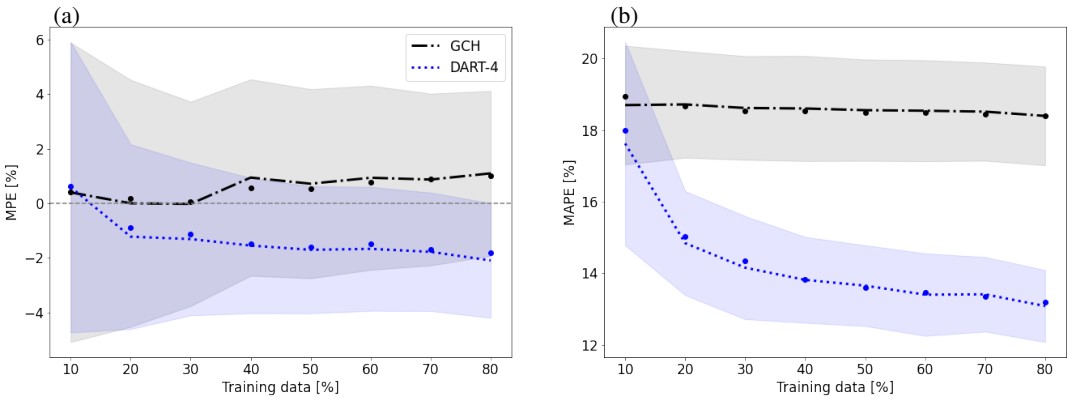

**Figure 14.** MPE (a) and MAPE (b) as function of training data size. Markers indicate the means ($\overline{\text{MPE}}$ and $\overline{\text{MAPE}}$), lines indicate the median and shaded areas indicate the standard deviation, corresponding to the fitted normal distributions in Fig. 11.

Fig. 8) are considered. Although the variability found in the field is not fully captured by either model, it is clear that the wake curl is better reproduced by DART than by GCH.

### 4.2.4 Dependency of performance on data set size

An important aspect of data-driven models is understanding how the amount of training data affects the model's accuracy. This
is especially relevant, as one of the most named drawbacks of data-driven models is their high need of data. This section studies the sensitivity of the accuracy of DART-4 to the amount of training data. DART-7 is not considered due to its long training time. Additionally, GCH is included in the analysis as it contains tuning parameters which could benefit from being tuned to a larger data set. All models are trained with a part of the full data set, ranging from 10 % to 80 %, and tested on the remaining 20 %, analogous to the procedure described in Sect. 3.3. Regardless of the amount of training data, the testing data are always
20 % of the original data set and consist of the same cases for fair comparison. When using e.g., 40 % of the data for training and 20 % for testing, the remaining 40 % is not used at all.

Figure 14 displays the accuracy of the models as a function of the size of the training data set. The metrics used for this analysis are again the distribution of MPE and MAPE values of the 96 resamples, corresponding to the normal distributions shown in Fig. 11. Figure 14a again reveals that DART-4 has a negative bias (MPE < 0 %), which is present regardless of the amount of
training data, whereas GCH typically has a small positive bias (MPE > 0 %). The uncertainty bands, representing one standard deviation indicated by the shaded area, are larger for GCH than for DART-4, while for both models the uncertainty is reduced when trained with more data.

Figure 14b displays the distribution of MAPE values of the 96 resamples as function of the data set size. GCH and DART-4 have a similar accuracy when few data are available, but DART-4 already outperforms GCH when as little as 20 % of the
data set ($\approx$ 75 cases or 13 hours) is used for training. Note that this does not indicate 13 hours of consecutive measurements, but rather 75 cases covering a range of meteorological conditions representative for the variability experienced by the turbine.

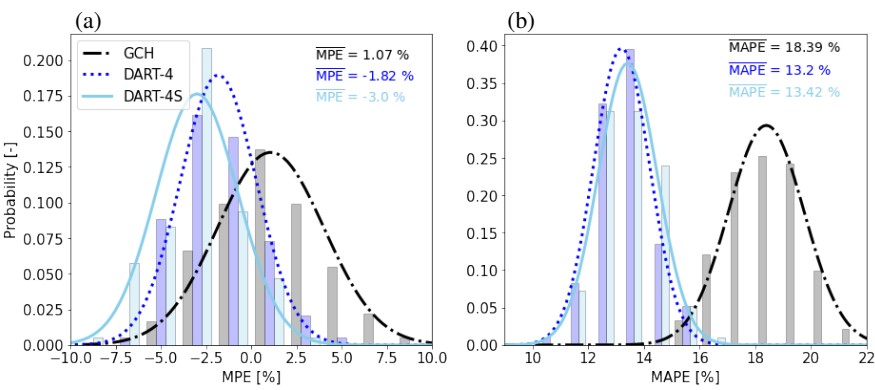

**Figure 15.** Same as Fig. 11 b-c, but with DART-4S (light blue).

Additionally, the accuracy of DART-4 seems to continue to improve when adding more data, albeit at a slower rate, whereas GCH hardly shows any improvement with higher data availability.

### 4.2.5 Performance with SCADA data as input

In this section, DART-4 is trained using only data routinely available to the operator (SCADA data) as input variables, called DART-4S. However, it still uses the measurements from the nacelle-mounted lidar to obtain the wake characteristics. Input variables includes power $P$, rotor speed $\omega$, wind speed $U_\mathrm{S}$ and turbulence intensity $\mathrm{TI_S}$ estimated from the cup anemometer and yaw misalignment $\phi_\mathrm{S}$ extracted from the wind vane. As discussed according to Fig. 4, this signal contains a systematic bias (Fleming et al., 2021) and is disturbed by the turbine yaw misalignment (Kragh and Fleming, 2012; Simley et al., 2021a), resulting in misalignments being overestimated by the nacelle vane. No correction was applied here, as data-driven models can compensate for any systematic biases. A similar reasoning can be applied to the use of $\mathrm{TI_S}$: although turbulence intensity estimates from a nacelle cup anemometer are affected by the rotor (e.g., Barthelmie et al., 2007), biases can be handled by data-driven models and are therefore acceptable. Additionally, data-driven models are better able to deal with noise than analytical models, which assume that the inflow information is undisturbed or need error terms inserted in the model equations (Schreiber et al., 2020).

The main issue of only using SCADA data is that there is no reliable estimate for the vertical wind speed profile. In this study, the shear $\alpha$ measured at the met mast is estimated from $\mathrm{TI_S}$ using the fitted linear relation: $\hat{\alpha} = 0.625 - 0.023\ \mathrm{TI_S}$. Because $\alpha$ and $\mathrm{TI_S}$ are quite weakly correlated (R = -0.47), this simple approach introduces uncertainty. However, developing a more sophisticated solution was deemed out of the scope of this work and this approach is deemed sufficient for the current purpose. An alternative approach could be to use strain measurements from the turbine's blades to estimate shear, as demonstrated in (Bertelè et al., 2017, 2021), although this would also involve additional sensors.

Figure 15 displays the results of DART-4S, using ($\phi_\mathrm{S}$, $\mathrm{TI_S}$, $U_\mathrm{S}$, $P$) as input, in a comparison with GCH and DART-4, both trained with met mast data. The accuracy of DART-4S is very similar to DART-4, showing a larger negative $\overline{\mathrm{MPE}}$ but an almost

identical $\overline{\mathrm{MAPE}}$. This indicates that using an arguably lower quality data set hardly affects the accuracy of the wake estimates. It is hypothesized that this is because the SCADA data better capture the atmospheric conditions at the turbine, whereas met mast data are subject to heterogeneity between met mast and turbine. This would counter the lower quality of the data, leading to only a slight decrease of the model accuracy. Interestingly, DART-4S is significantly more accurate than GCH, even though the latter needs undisturbed measurements as input.

## 5 Discussion

Section 5.1 discusses the measurement campaign and its accuracy. In Sect. 5.2 the limitations of the data-driven model are reviewed. Finally, Sect. 5.3 focuses on the implication of this study's results for future work.

### 5.1 Campaign

This section discusses some key takeaways for future campaigns (Sect. 5.1.1) and an uncertainty analysis considering measurement errors (Sect. 5.1.2).

#### 5.1.1 Lessons learned

Several considerations regarding the experimental campaign are noteworthy. First, the nacelle-mounted lidar's scanning strategy was based on Brugger et al. (2019, 2020) and evaluated systematically using large eddy simulation results and a lidar simulator. However, during this analysis, data losses were not considered. Subsequently, in the field data occasionally all information at one height was filtered out, leaving only information at four heights for the analysis (Sect. 2.3), which could lead to interpolation errors. A more robust approach would have been to perform seven instead of five consecutive PPI scans, although the accuracy of the wake reconstruction method is slightly lower (see Fig. 2). Lastly, in this study only one distance of $4D$ was targeted, but for other purposes it could be desirable to target multiple positions at once. This would likely require more PPI scans with a larger range of elevation angles, as used in (Brugger et al., 2020). Large elevation angles are needed to capture the wake close to the turbine, whereas small elevation angles capture the wake further downstream.

Further, no systematic hard target analysis was performed with the nacelle based lidar. The horizontal offset relative to the turbine's center axis could be estimated from a set of coarser PPI scans, but no vertical offset could be estimated. Although this is not expected to have large influence on the results presented here, as is also discussed in Sect. 5.1.2, it is recommended to always carry out a hard-target analysis in future measurement campaigns to reduce the uncertainty of the measurements.

#### 5.1.2 Measurement uncertainty

Although the measurement data after filtering have been considered as the "ground truth" in this study, a few aspects affecting the data quality should be considered. Homogeneity of the background flow is assumed, as well as a vertical wind profile that can be described with the power law, which is not always satisfied. This refers specifically to the trees in the wind direction sector around $\delta = 350°$ that are assumed not to affect met mast data, as mentioned in Sect. 2.1. Besides, turbines T3 (induction

**Table 3.** Overview of all tests carried out for the uncertainty analysis.

| Test | Description |
|------|-------------|
| 0 | Original; Fig. 11 |
| 1 | $\phi_{\text{PPI}} - 2.18°$ |
| 2 | $\phi_{\text{PPI}} - 1.09°$ |
| 3 | $\phi_{\text{PPI}} + 1.09°$ |
| 4 | $U_{\text{h}} + 0.2 \text{ m s}^{-1}$ & $U_{\text{lt}}$ - $0.2 \text{ m s}^{-1}$ |
| 5 | $U_{\text{h}}$ - $0.2 \text{ m s}^{-1}$ & $U_{\text{lt}} + 0.2 \text{ m s}^{-1}$ |
| 6 | $\delta_{\text{h}} + 1.5°$ & $\delta_{\text{lt}}$ - $1.5°$ |
| 7 | $\delta_{\text{h}}$ - $1.5°$ & $\delta_{\text{lt}} + 1.5°$ |

zone) and T4 (wake) are assumed to not affect the wake, although this cannot be ruled out entirely. Lastly, the lidar measurements are inherently subject to probe volume averaging and a different filtering method than the one described in Sect. 2.3 will retain other information and therefore result in slightly different wake characteristics.

Additional analyses were carried out to investigate the effect of measurement uncertainty on the results presented in Sect. 4, specifically those in Fig. 11. An overview of these tests is displayed in Table 3. First, the influence of the missing hard target analysis (Sect. 5.1.1) is investigated. In the original measurements, an upward vertical displacement of the wake center of

$0.15D$ was observed, averaged over all 382 cases. Although displacements of this magnitude have been observed in numerical simulations (Sengers et al., 2020), it is here assumed that this is purely the consequence of an installation error of the lidar. Such a displacement at $4D$ downstream would come from a downward angle of $\tan^{-1}(0.15/4) = -2.18°$. In test 1, the elevation angles of all lidar scans were adjusted with this value, resulting in an average vertical wake center displacement of zero. Since vertical wake center displacements have been observed in other studies (e.g., Bastankhah and Porté-Agel, 2016; Sengers et al.,

2020), the previous test was not deemed completely realistic. Additional tests (2 and 3) with half the correction value, as well as a positive correction value, were carried out. The next set of tests varies the wind speed and direction measured at the met mast. As noted in Sect. 2.5, the accuracy of the anemometer is $0.2 \text{ m s}^{-1}$ and the accuracy of the vane is $1.5°$. These values are used in tests 4-7, varying the measurements at hub height ($_{\text{h}}$) and lower tip height ($_{\text{lt}}$) in opposite direction, investigating the maximum influence of these uncertainties. Note that this does affect the shear and veer as well.

In all tests, DART-4 (DART-3, DART-7 and DART-4S omitted for brevity) and GCH are trained on the adjusted data for all 96 resembles. Their accuracy on the testing data is evaluated using $\overline{\text{MPE}}$ and $\overline{\text{MAPE}}$, and shown in Fig. 16. Compared to the original (Test 0) of Fig. 11, Test 1 shows a slightly poorer performance of DART-4 (larger $\overline{\text{MPE}}$ and $\overline{\text{MAPE}}$). It is hypothesized that this is due to the fact that now the 5 PPI scans do not fully target the rotor area anymore, resulting in less relevant information about the wake to estimate the available power. GCH on the other hand, seems to perform better

(smaller $\overline{\text{MPE}}$ and $\overline{\text{MAPE}}$) than the original. Since no vertical displacement is estimated with GCH, this new data more closely resembles to the model's assumptions. Tests 2 and 3 confirm this, as GCH's $\overline{\text{MAPE}}$ increases with the magnitude of the vertical wake center displacement.

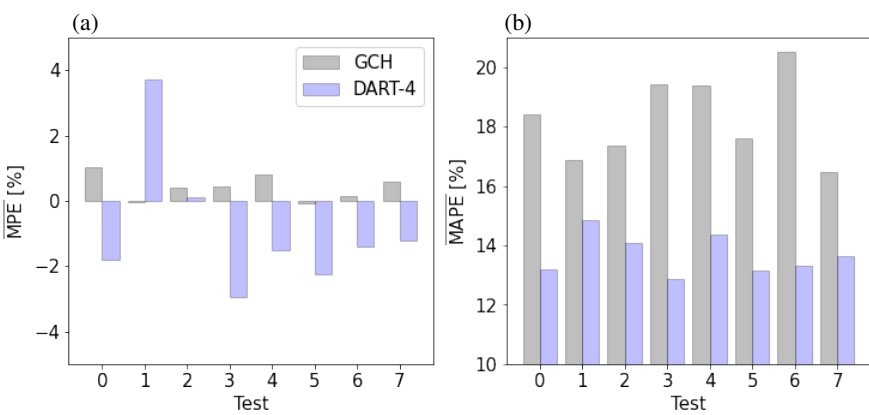

**Figure 16.** Results of the tests of Table 3. $\overline{\mathrm{MPE}}$ and $\overline{\mathrm{MAPE}}$ of Test 0 correspond to the values found in Fig. 11.

Tests 4 and 5 illustrate that both models perform better when the shear is decreased (Test 5) compared to when the shear is increased (Test 4), which relates to the fact that more uniform conditions are easier to reproduce. Lastly, GCH performs worse in Test 6 and better in Test 7 compared to the original, but no satisfying explanation was found.

In general, these tests demonstrate that the more closely a data set resembles the model assumptions, the better the model performs. DART-4 typically shows a higher $\overline{\mathrm{MPE}}$ and a lower $\overline{\mathrm{MAPE}}$ compared to GCH, which is similar to the results presented in Sect. 4. This uncertainty analysis is believed to demonstrate that the results presented in this study are robust and not very sensitive to measurement uncertainty.

## 5.2 Data-driven model

DART's quantitative results presented in this study are not fully generalizable. The fitted coefficients in Eq. (3) are only valid for the scenario considered in this study and it is unknown how the model's accuracy transfers to different scenarios, such as other turbine types and downstream distances. Besides, while the range of achieved yaw misalignments is typical for field experiments nowadays (e.g., Fleming et al., 2020, 2021; Doekemeijer et al., 2021), future campaigns could see larger misalignments such as those currently considered in numerical studies (e.g., Howland et al., 2016; Martínez-Tossas et al., 2019; Bastankhah et al., 2022). Further lidar measurements would be needed in new scenarios to guarantee accurate model estimates, and although it needs relatively few data to retrained in new situations (Sect. 4.2.4), this limits the potential for application of data-driven models in wake steering control.

To overcome this, future work should attempt to generalize the model's coefficients. Alternatively, model equations using coefficients determined with previous numerical or experimental data could still be used at new locations to generate a first estimate of the wake characteristics. Assuming that the wake position and shape are sufficiently accurately modeled, coefficients for the wake deficit could be retrained using SCADA data by deducing a rotor equivalent wind speed.

Lastly, other data-driven models could be used. Currently, to the best of our knowledge DART is the only data-driven wake model available that does not make use of complex black-box models such as neural networks. Although it would be inter-

esting to compare different data-driven models, more complex models typically need more data. For instance, Asmuth and Korb (2022) proposed a neural network and showed they need at least 800 cases to train the model for non-yawed cases only. Although their results are extremely promising, extending this to include wake steering would likely require a substantially longer measurement campaign.

## 5.3 Implications for future work

As noted in the introduction, the industry appears to be hesitant to adopt the wake steering strategy due to large uncertainties. To overcome this, yaw controllers need to become more sophisticated, for instance by using closed-loop controllers (Doekemeijer et al., 2020; Howland et al., 2020) or using preview information (Simley et al., 2021b). On the other hand, the low-fidelity wake models that are utilized to determine the yaw misalignment set points used by the yaw controller need to become more accurate.

This study contributes to the latter by showing that both DART and GCH perform well on average (small systematic bias), but that DART can capture a higher degree of variability observed in the field. Besides more accurate estimations of the wake deficit, which historically has been the main focus of wake models, this extends to other wake characteristics like wake curl and wake center location. The latter is especially important for wake steering, as erroneous steering can steer the wake into a downstream turbine.

Since DART shows a higher accuracy than GCH in estimating wake characteristics, it can be hypothesized that when using DART to determine yaw misalignment set points used by the yaw controller, the wake steering strategy can be applied more successfully. This can consist of achieving higher power gains when wake steering is performed successfully, or reducing power losses due to erroneous steering. However, an extensive campaign would be needed to investigate this, which was considered out of the scope of the current work.

On a more general level, this study shows that data-driven models are a viable alternative to analytical models. Whereas data-driven models have often been criticized for their complex nature, this study has demonstrated that accurate estimations can also be obtained with a very simple linear model.

While the current model focuses on estimated wind speed and consequently power, a similar methodology could be developed to estimate turbulence and consequently turbine loads. Alternatively, it would be interesting to combine analytical and data-
driven models in hybrid models. Such models could initially benefit from the robustness of analytical models, but exploit the higher accuracy of data-driven models when more data becomes available.

## 6 Conclusions

This study uses nacelle-based lidar measurements of the wake of a commercial turbine with a fixed intentional yaw misalignment. Performing a trajectory of five consecutive Plan Position Indicator (PPI) scans with different elevation angles, a vertical
wake cross-section at four rotor diameters downstream is reconstructed. Utilizing the Multiple 1D Gaussian method, wake characteristics are obtained. The lateral wake center displacement and wake curl observed in the field compare well with large

eddy simulation results. The results from the lidar measurements demonstrate the occurrence of the wake curl in the field, which had not conclusively been shown in literature before. This is due to small curling observed for yaw misalignments below 20°, which could be missed when using a different scanning trajectory.

The field measurements are subsequently used to train and validate the DART model, and compare the accuracy of the trained data-driven model to the accuracy of the GCH model. When estimating the observed wake characteristics with both wake models, it is demonstrated that DART systematically outperforms GCH. Depending on the number of input variables used for DART, the error is reduced by between 19 % and 36 % compared to GCH. The metric used here is the Mean Absolute Percentage Error of the available power of a virtual downstream turbine, averaged over 96 resampled testing data sets. Espe-

cially when the turbine is misaligned or high vertical shear is observed, DART outperforms GCH. Besides, DART requires a relatively small amount of training data (about 75 cases at specific set points) to outperform GCH. Further analysis suggests that DART's accuracy is hardly affected when only considering SCADA data as input in comparison to using undisturbed measurements from a met mast.

DART shows a high accuracy in the current study, targeting a downstream distance of four rotor diameters and using a range

of yaw misalignments commonly used in field experiments. However, these results cannot directly be generalized and further lidar measurements are needed to retrain DART for new scenarios, limiting its applicability. Regardless, this study's results are believed to demonstrate the potential of data-driven wake models and the role they can play in the further deployment of wake steering control strategies.

*Code and data availability.* The Data-driven wAke steeRing surrogaTe model (DART), including a short tutorial, is available for download

at: https://github.com/LuukSengers/DART (https://doi.org/10.5281/zenodo.7442225, Sengers and Zech (2022)).

The large eddy simulation results cannot be shared due to confidentiality of the turbine specific aerodynamic characteristics.

A selection of measurement data are available at https://doi.org/10.5281/zenodo.7741395 (Sengers, 2023). This data subset provides ten-minute averaged input parameters and lidar scans of cases with $U_\mathrm{h} \approx 8$ m s$^{-1}$.

## Appendix A: Lists of Abbreviations and Symbols

*Abbreviations*

| | |
|---|---|
| APE | Absolute Percentage Error |
| CNR | Carrier-to-noise ratio |
| DART | Data-driven wAke steeRing surrogaTe model |
| EC | Eddy Covariance |
| GCH | Gaussian-Curl Hybrid model |
| LES | Large eddy simulation |
| LOS | Line-of-sight velocity |
| LUT | Look-up table |
| MAE | Mean absolute error |
| MAPE | Mean absolute percentage error |
| $\overline{\text{MAPE}}$ | Mean of the mean absolute percentage errors of all resamples |
| MM | Meteorological mast |
| MPE | Mean percentage error |
| $\overline{\text{MPE}}$ | Mean of the mean percentage errors of all resamples |
| PALM | PArallelized Large-eddy simulation Model |
| PE | Percentage error |
| PPI | Plan position indicator |
| R | Pearson correlation coefficient |
| SCADA | Standard supervisory control and data acquisition |
| VAD | Velocity-azimuth display |

*Symbols*

| | |
|---|---|
| $\alpha$ | Shear |
| $\alpha_{\text{GCH}}$ | Tuning parameter GCH |
| $\beta_{\text{GCH}}$ | Tuning parameter GCH |
| $\delta$ | Wind direction |
| $\delta_{\text{h}}$ | Wind direction at hub height, measured at MM |
| $\delta_{\text{lt}}$ | Wind direction at lower tip height |
| $\delta_{\text{S}}$ | Wind direction at hub height, measured at nacelle (SCADA) |
| $\delta\alpha$ | Veer |

| | |
|---|---|
| $\mu_y$ | Lateral wake center displacement |
| $\mu_z$ | Vertical wake center displacement |
| $\omega$ | Rotor speed |
| $\omega_{\mathrm{lidar}}$ | Angular speed of lidar scan |
| $\phi$ | Yaw misalignment angle |
| $\phi_{\mathrm{PPI}}$ | Elevation angle PPI scans |
| $\phi_{\mathrm{S}}$ | Yaw misalignment angle, measured at nacelle (SCADA) |
| $\phi_{\mathrm{t}}$ | Target yaw misalignment angle |
| $\phi_{\mathrm{VAD}}$ | Elevation angle VAD lidar |
| $\sigma_y$ | Width of wake at center height |
| $\sigma_z$ | Vertical extent of wake |
| $\theta_{\mathrm{VAD}}$ | Azimuth angle VAD lidar |
| $A_z$ | Amplitude of wake deficit normalized with $U_{\mathrm{h}}$ |
| $C_{\mathrm{T}}$ | Thrust coefficient |
| curl | Wake curl |
| $D$ | Rotor diameter |
| $k_{a,\mathrm{GCH}}$ | Tuning parameter GCH |
| $k_{b,\mathrm{GCH}}$ | Tuning parameter GCH |
| $P$ | Power |
| $P_{\mathrm{av}}$ | Available power |
| $s_a$ | Quadratic wake width parameter |
| $s_b$ | Linear wake width parameter |
| TI | Turbulence intensity |
| tilt | Wake tilt |
| $\mathrm{TI_S}$ | Turbulence intensity, measured at nacelle |
| $U$ | Wind speed |
| $U_{\mathrm{def}}$ | Wind speed deficit |
| $U_{\mathrm{eq}}$ | Rotor equivalent wind speed |
| $U_{\mathrm{h}}$ | Wind speed at hub height. When indicating measurement data, it is measured at MM |
| $U_{\mathrm{lt}}$ | Wind speed at lower tip hub height |
| $U_{\mathrm{S}}$ | Wind speed at hub height, measured at nacelle (SCADA) |

*Author contributions.* BAMS designed the experiment, processed the data, generated the results, and wrote and edited the manuscript. GS provided intensive consultation on the experimental design and generation of the results. PH was heavily involved in organizing the campaign and contributed to the processing of the raw data, as well as translated the aerodynamic properties of the turbine provided by the operator in Bladed into FAST. MK provided general consultation and had a supervisory function. All coauthors reviewed the manuscript.

*Competing interests.* The authors declare that they have no conflict of interest.

*Acknowledgements.* The authors would like express their gratitude to eno energy GmbH, specifically to Alexander Gerds, for the opportunity to carry out these experiment on one of their commercial turbines. Carlo Sucameli and Vlaho Petrović are acknowledged for the implementation of the controller, and Stephan Stone, Anantha Padmanabhan Kidambi Sekar and Jörge Schneemann are thanked for helping during the installation of the sensors. Hauke Beck is recognized for the discussions on the filtering of lidar data. Matthias Zech is thanked for the discussions on the results obtained with DART and Marijn Floris van Dooren is acknowledged for his help with LiXim. Paul van der

Laan and Andreas Rott are thanked for general discussions. The presented work has been carried out within the national research project "CompactWind II" (FKZ 0325492H), funded by the Federal Ministry for Economic Affairs and Energy (BMWi) on the basis of a decision by the German Bundestag. Computer resources have been provided by the national research project "Heterogener Hochleistungsrechner für windenergierelevante Meteorologie- und Strömungsberechnungen (WIMS-Cluster)" (FKZ 0324005), funded by the Federal Ministry for Economic Affairs and Energy (BMWi) on the basis of a decision by the German Bundestag.

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
