# Peer review of "Validation of an interpretable data-driven wake model using lidar measurements from a free-field wake steering experiment"

_Wind Energy Science, 2022_

## Referee Comment (RC3)

**Review of Manuscript "Validation of an interpretable data-driven wake model using lidar measurements from a free-field wake steering experiment" by Sengers et al. (wes-2022-118)**

The manuscript by Sengers et al. presents the (free-field experimental) validation of the DART model which was introduced by the authors before. Model predictions with a focus on wake steering of a single turbine are compared with both the experimental data, comprising measurements of two scanning lidars and the turbine data, and the Gaussian-Curl Hybrid (GDH) model as reference.

Overall, the manuscript meets its defined objectives and describes the findings in a well comprehensible manner. However, I have identified three main deficits, I will briefly summarize (as major issues), followed by a list of minor points which should be addressed before a publication of the manuscript.

Major issue #1: As I understand it, the manuscript contrasts from the earlier publication(s) by using free-field data for validation. The used measurement campaign has however, as stated by the authors, several shortcomings. These are described but their impact on the results is, in my view, not sufficiently discussed and quantified, respectively. A more detailed uncertainty quantification, elaborating on these impacts, may be very useful to underline the findings of the study.

Major issue #2: The main findings of the presented study focus on the accuracy of wake characteristics and the comparison to results of the reference model. What I am missing is a more detailed discussed how these findings impact the possible application of wake steering in terms of the introduced control strategies in wind farms. I suggest to add another sub-section in the Discussion (section 5) which addresses this, the implications (qualitative and quantitative) on an application. In addition to seeing the need for this discussion, I also think the current section 5 is rather short and should be elaborated on.

Major issue #3: Though clearly written, I think the presentation and first of all the structure of the manuscript should be improved. Section order / levels and titles should be revised and in some cases made more consistent. More concrete suggestions in the list below.

Minor issues (in order of their appearance in manuscript):

l. 26 – "test turbines" instead of "testing turbines"

l. 27 – Referring to "simulations and wind tunnel experiments", I think it would be helpful to shortly describe in some more detail related pros and cons (of these) in contrast to a free-field experiment.

ll. 29 – Please describe in one more sentence how "erroneous yawing" may interact with wake steering more explicitly.

ll. 73 – Do we need explicit wake steering for such an experiment or is it just about the variation of yaw angles? Please comment on this.

ll. 75 – How do you define the difference between "validations and comparisons"? Please comment.

l. 79 – What does "This" refer to?

ll. 79 – I suggest to use commas for this list.

l. 84 – Please write a short introduction to this section, also introducing the sub-sections.

ll. 94/95 – Can you elaborate on this, why only a fixed yaw offset in this sector?

ll. 112 – I do not think this title fits optimally to the other titles. Again, I also suggest to add a short introduction to the following sub-sub-sections. And why do you not use numbering (2.2.1, ..) for the following paragraphs?

ll. 114 – This sentence ("A pulsed ..") needs to be rewritten. Please check again what you want to say.

l. 115 – "horizontal plane" is only true if you use a single PPI with zero elevation – please be more explicit here.

Figure 1 – The colour scheme is not optimal here (in particular, as you are not showing the colour bar as legend). Could you use less blue?

l. 137 – Please detail the link between $U_{eq}$ and $P_{av}$.

l. 138 – Delete one "with".

l. 139 – Why do you give the equation for PE but not APE here? This is rather confusing as you write about APE before.

l. 140 – "reconstruction" of what?

l. 142 – ".. were not tested, as this would remove .." This is only true if you require a symmetry. Please comment on this.

l. 145 – "hold" instead of "holds"

Figure 3 – In my print it is not really "yellow" – please check, and maybe use another colour.

Figure 3 – Why was the "second cluster [..] omitted"? I think this is neither sufficiently described in the figure caption nor in the main text.

l. 167 – Please explain briefly why you have "slightly different azimuth angles".

l. 174 – Please mention here again that the ground-based lidar is also a scanning lidar of type 200s.

l. 187 – Here you could introduce the abbreviation "met".

Figure 5 (and others) – Difference symbols are used for veer in figures and text – "del" and "delta", respectively.

l. 239 – Again, I suggest to add some introduction text to the section.

l. 251 – "their" instead of "it's"

l. 278 – There is only one sub-level (3.1.1) – please revise section structure.

l. 300 – I believe this should say "for this particular experiment" rather than "in the free field".

Figure 8 – As above, use "delta alpha" as in main text.

Figure 10 – Please introduce the colour code for (a).

Figure 11 – The "yellow" text is very difficult to read – please select another colour.

l. 437 – There should be some introduction to the following sub-sub-sections.

l. 444 – "20%" of the original dataset – suggest to add this detail here.

l. 480 – Add some introduction to the section here.

ll. 480 – As pointed out above, this section should be elaborated on. Currently it only addresses the limitation.  I suggest a more in-depth discussion of the application here.

Appendix A – I do not think this appendix is really needed.

---

## Author Comment (AC1)

**Authors' response to "Validation of an interpretable data-driven wake model using lidar measurements from a free-field wake steering experiment"**

Balthazar Sengers[1,2], Gerald Steinfeld[1], Paul Hulsman[1], Martin Kühn[1]
[1]ForWind, Institute of Physics, Carl von Ossietzky University Oldenburg, Küpkersweg 70, 26129 Oldenburg, Germany
[2]Current affiliation: Fraunhofer IWES, Küpkersweg 70, 26129 Oldenburg, Germany
Correspondence: balthazar.sengers@uni-oldenburg.de

**Response to all reviewers**
The authors appreciate the feedback from all reviewers and believe that the manuscript has been improved after implementing the reviewers' comments.
As suggestion by Reviewer 2 (technical correction 3), the notion "free field" should be replaced by "field", acknowledging that there are external influences that affect the collected data. This results in a small change in the title.
Lastly, some sentences have been rephrased to minimize self-plagiarism, but these additional changes have not been included in the author's response.

The author's response to each of the reviewers' comments (in black) can be found below (in red), as well as the rephrased sentences or added text (in blue).

**Reviewer 1**
General comments
Field measurements of a nacelle mounted Doppler lidar are used to train and validate a data-driven wake model for yawed wind turbines in comparison to an analytical wake model.
The research question is relevant to the field of wind energy. The descriptions of the measurement campaign and the methods are missing some information. More consideration should be given to the assumptions of the analytical model and how they might affect its results. The presentation of the results is good and the conclusions drawn are reasonable. I believe my comments can be addressed by the authors and then I would recommend accepting the manuscript.

Main comments
1. One of the conclusions is that the data-driven wake model performs better than the analytical wake model under high wind shear or yaw misalignment of the wind turbine. The analytical model was designed for the far-wake, but the measurements are taken at x=4D, which might be still within the near-wake for some conditions. Therefore, I believe the authors should make an effort to investigate if the cases with large errors of the analytical model might also be linked to applying the analytical model to the near wake for which it is not designed (see specific comments for lines 342-344).
This comment is addressed in specific comment 15.

2. The observed upward displacement of the wake center is crucially dependent on the lidar's orientation (especially the tilt). However, the manuscript does not provide any information on the accuracy of the tilt/pitch/roll sensors used to correct the lidar measurements nor how they were corrected exactly (see specific comments for lines 200-204 and 360-363).
This comment is addressed in specific comments 8 and 18.

Specific comments

1. Line 71/72: Provide a citation for the model comparison with LES.

It is now clarified that the comparison with LES was also performed in Sengers et al. (2022)

"In a comparison using large eddy simulation (LES) results, Sengers et al. (2022) demonstrated that DART outperformed the Gaussian and GCH models, especially under stable atmospheric conditions."

2. Line 90: Provide more details on the lidar used.

The serial numbers of the lidars have been added.

"The nacelle of turbine T1 was equipped with a downstream facing *Leosphere Windcube 200S* (serial no. WLS200S-024) pulsed lidar (Sect. 2.3)."

"A second pulsed lidar of the same type (serial no. WLS200S-023) was installed west of the turbine to measure inflow profiles (Sect. 2.4)."

Additionally, more information on the scanning strategy has been added in line 151 of the original manuscript.

"The range gate length was set to 25 m, corresponding to a pulse duration of 100 ns. Range gates were defined between 50 m and 1340 m with 5 m spacing. However, in the processing phase only data up to 820 m were used to avoid the influence of the ground in the PPI scan with the lowest elevation angle."

Similar information has been added to Sect. 2.3 (now Sect. 2.4) on the VAD lidar.

"Also for this lidar, the range gate length was set to 25 m, corresponding to a pulse duration of 100 ns. Range gates were defined between 50 m and 840 m with 5 m spacing."

3. Line 110: What was the surface coverage at the measurement site? Are there buildings, trees, forests, or other features that might affect the flow at the height of the measurements?

More details on the surroundings have now been added.

"Two villages with low buildings were located about 1 km from T1, directly upstream for wind directions around $\delta = 265°$ and $\delta = 320°$, mainly outside of the studied wind direction sectors. The dominant vegetation in the area is of agricultural nature, with patches of trees and bushes between the fields. These trees could affect the measurements for $\delta \approx 350°$, as noted in Hulsman et al. (2022) using data from the same site. This influence was accepted, as omitting this sector would result in large data losses."

4. Line 134: I believe "time / temporal averages" might be a more clear description than "point-wise averages".

This has been changed in line 134, as well as in line 168 (of original manuscript).

"Temporal averages were taken for all points in the scanning cycle." (line 134)

"Next, the scans were temporally averaged as long as not more than two data points within a 10-minute window were missing." (line 168)

5. Line 149-151: Stating the elevation angles explicitly would make it easier to get an overview of the lidar scans (or a schematic of the scan).

The authors agree that this information was indeed missing and has now been added.

"The elevation angles ($\varphi_{PPI}$) of these scans were (-7.0°, -3.5°, 0.0°, 3.5°, 7.0°), with the outermost scans targeting upper and lower tip height at $4D$."

6. Line 169: For consistent language to the previous text consider using "PPI scan" instead of "scan" and "10-minute window" instead of "data set".

The authors appreciate this detailed comment on consistency. Besides this line, "scan(s)" has been changed into "PPI scan(s)" at multiple locations in the text (not listed here). Changing "data set" into

"10-minute window" was only done in this line, as in the rest of the manuscript "data set" refers to the collection of data, rather than just one case.
"When more than 25 % of the measurements were filtered out, as is the case with Figure 3b, the averaged PPI scan was removed from the 10-minute window, resulting in fewer than five PPI scans."

7. Line 179: It is not clear to me what the maximum would be here.

This sentence was rewritten.
"The measured data points were binned by their LOS and CNR values and the number of data point in each bin were counted. Bins having a count less than 10 % of the bin with the highest count were omitted."

8. Line 200-204: What was the accuracy of the tilt and pitch provided by the GPS sensor? And what was the temporal resolution?

The author agree that this information was missing. More details on the system have been added. The device's manual states a RMSE of less than 0.1°. How this translates to an uncertainty at $4D$ was estimated using the rules of error propagation:

$$\Delta y = \frac{\partial \; 4D\tan(\phi_{\mathrm{PPI}}\frac{\pi}{180})}{\partial \; \phi_{\mathrm{PPI}}}$$

$$\Delta y = 4D\frac{1}{\cos^2(\phi_{\mathrm{PPI}}\frac{\pi}{180})}\; \frac{\pi}{180}\Delta\phi_{\mathrm{PPI}}$$

For an elevation angle of $\phi_{PPI} = 0°$, using an error of $\Delta\phi_{PPI} = 0.1°$, this results in a error of the vertical position of $\Delta y = 0.88$ m.
"Additionally, the nacelle of T1 was equipped with a *Trimbl SPS* three-antenna GNSS system to measure orientation, roll and tilt. This system was operated at a sampling frequency of 10 Hz and its measurements have a Root Mean Square Error of less than 0.1°. This results in a spatial error of less than 1 m at $4D$ downstream."

Lastly, how were the lidar scans corrected exactly? Was each measurement point corrected individually or was the correction applied on average for a PPI / a cycle of five PPIs / a 10-minute window?

The authors agree that this information was missing and has now been added.
"Orientation measurements, averaged to 10-minute values to smooth out high-frequency vibrations, were used to compute the yaw misalignment $\phi$ of the turbine relative to the wind direction $\delta_h$ measured at the met mast. These measurements were then used to correct the PPI scans' azimuth angles. Likewise, 10-minute averaged nacelle tilt angles were used to correct the PPI scans' elevation angles, but the scans were not corrected for roll as it was expected to only have a small influence on the results."
For clarity, this information has also been added to line 171 of the original manuscript:
"Lastly, the PPI scans' azimuth and elevation angles were corrected with the nacelle's 10-minute averaged tilt angle and misalignment (see Sect. 2.7)."

9. Line 232-234: The medians given in the text and the medians given in Fig. 4 are different from (I assume that mu is the median).

The authors regret having overlooked this inconsistency. Small changes were made to the processing of the data between the initial drafting and submission of the manuscript, resulting in slightly different

values for these medians. It was checked and confirmed that the values indicated in the figure are the correct ones.

"For greedy control, the median shows a small bias of $\phi$ = -0.94°, suggesting a calibration error of the nacelle's wind vane. For a target angle $\phi_t$ = +15°, the median achieved $\phi$ = +11.14°, whereas for $\phi_t$ = -15°, $\phi$ = -13.19° is achieved."

10. Line 236: I assume "normal" means here that the values fall within the typical range for the ABL and not the normal distribution in the statistical sense (maybe rephrase).

This sentence has been rephrased.

"… all showing values in a range that is physically reasonable."

**11.** Line 245-251: It seems unintuitive to me that the quantification of the wake characteristics from the lidar measurements is described in the section of the data driven model.

This has now been moved to its own Sect. 3.1. Additionally, some information on the composition method has been added to this section.

"The Multiple 1D Gaussian method (Sengers et al., 2020) tis utilized to obtain quantifiable wake characteristics, listed in Table 1. This method fits a 1D Gaussian through the wake deficit data normalized by the wind speed at hub height ($U_{def}/U_h$) in the horizontal plane for every height level, in the current study obtained from five consecutive PPI scans. This results in a set of local wake deficits (amplitude), center positions (location) and widths (standard deviation) for each height. By fitting another 1D Gaussian through the set of local deficits in the vertical, the vertical deficit profile can be determined. The position of the maximum deficit in this profile is then considered as the vertical position of the wake center. The horizontal position of the wake center is determined by interpolating the set of local center positions to this height. A second-order polynomial is fit through the set of local wake center positions to find the wake curl and tilt. The same method is applied for the wake widths to find their profile as function of height.

The reverse of this method, hereafter called composition method, can be used to obtain a vertical cross-section of the wake from a set of wake characteristics. For more details on the Multiple 1D Gaussian method and the composition method, the reader is referred to Sengers et al. (2020, 2022)."

12. Line 249: The azimuth-opening angle of PPI scans is 70°, but the wake will take up only a small part of this window. Which of the vertical slices are selected for the vertical 1D Gaussian fit? (Or how are fits rejected, which are outside of the wake?)

The authors acknowledge that this information was missing and has now been added, see comment 15.

13. Line 329: Consider clarifying that the training time is the computation time of the training process (it could be misunderstood as the time window of the training data set).

The authors agree with this clarification. Similar changes have been made for DART-4 and DART-7.

"...hence the total computation time to determine the best set of input variables is approximately 1.5 hours." (line 329)

"...hence with 18 possible sets of input variables the total computation time needed for training is 18 hours." (line 336)

"The computation time needed to train DART-7 is approximately one month if not parallelized." (line 338)

14. Line 341: Can you provide details on the model physics (e.g. the model equations would be helpful)? The provided reference seems to focus on technical aspects of the implementation.

The authors acknowledge that a detailed description of the analytical model could be desirable. However, since the GCH model is only used as a reference (or benchmark) and no developments were

done to this model in this study, the authors believe it is not necessary to provide an elaborate description of the model here. The citation to the reference paper is now repeated, as well as a very short summary of the model's main details.
"The state-of-the-art GCH model (King et al., 2020) as available in version 3.0rc4 of the FLORIS framework (NREL, 2022) acts as a reference model in this study. The GCH model incorporates the spanwise and vertical velocity components (Martínez-Tossas et al., 2019) due to the present vortices to the Gaussian wake model (Bastankhah and Porté-Agel, 2014, 2016; Niayifar and Porté-Agel, 2016)."

15. Line 342-344: If using a far-wake model for the near-wake, one would expect to see bad results. The correlation between the observed velocity deficit and the Gaussian fit (which are already done for determining the wake characteristics as described in Sect. 3.1) could be used to identify if the conditions are met to apply the model. It should be possible to identify and remove 10-minute windows, which do not fulfill the model requirements. Alternatively, the correlation could provide a metric to investigate a dependency of large errors to a non-Gaussian velocity deficit in a manner similar to Fig. 12.

The authors appreciate this comment and the elaborate suggestion. The authors acknowledge that the GCH model was not developed for the near wake and might therefore show higher errors. However, the same can be said for the DART model, as this model also assumes that the wake deficit can be described as a Gaussian.

The procedure suggested by the reviewer was actually implemented and the authors regret that this detail was absent in the original manuscript. This check is the final step in the selection procedure described in Sect. 2.7 (now Sect. 2.8), and has been added there now, combined with Comment 12.

"Lastly, the 10-minute averaged cases were evaluated by the Multiple 1D Gaussian method (see Sect. 3.1). Since the opening angle of the PPI scans is 70°, it can be expected that wakes from other turbines are also visible in the measurements. To prevent using an incorrect wake, the scans are sliced around the expected location of the considered wake. Boundaries of these slices are determined by the maximum wind speeds between the scan's center, corrected for the yaw misalignment, and 150 m left and right of this center. Furthermore, the correlation coefficient (R) of the Gaussian fit with the wake deficit observations needed to be higher than 0.85 (empirically determined) to be considered, removing cases that do not fulfill the model assumptions of a Gaussian wake deficit."

16. Line 346: Have those tuning parameters physical meanings like a wake growth rate (since k is frequently used for this)?

Consistent with the Gaussian model, these parameters relate to the far-wake onset and the wake growth rate.

"The model tuning parameters ($\alpha_{\text{GCH}}$, $\beta_{\text{GCH}}$ for the far-wake onset, and $k_{a,\text{GCH}}$, $k_{b,\text{GCH}}$ for the wake growth rate) ..."

17. Line 352-353: Based on the description of the Multiple 1D Gaussian method, it is not yet clear to me how the vertical and horizontal displacement of the wake center are exactly determined. For example: for each 1D Gaussian fit along a PPI scan (horizontal), a lateral wake displacement is determined (five in total, if no PPI scan was rejected due to the CNR). Are they then averaged or is a specific one selected?

This has been clarified in the description of the Multiple 1D Gaussian method, see comment 11.

18. Line 360-363: This result for the vertical wake displacement is depends on the precise orientation of the lidar. If the wind turbine and subsequently the lidar are tilted backwards due to tower bending, the lidar data would suggest that the wake is displaced upwards. Bromm et al (2018) have a discussion on this issue. Are the pitch, roll, and tilt readings from the GPS sensors

accurate enough to exclude this issue here? If the leveling of the lidar is not an issue, it might be further interesting to test if the upward displacement of the wake center holds for both wind direction sectors to exclude topographical effects.

Bromm, M, Rott, A, Beck, H, Vollmer, L, Steinfeld, G, Kühn, M. Field investigation on the influence of yaw misalignment on the propagation of wind turbine wakes. Wind Energy. 2018; 21: 1011– 1028. https://doi.org/10.1002/we.2210.

The authors want to thank the reviewer for this suggestion. Each case has its own (10-minute averaged) tilt angle, that is obtained with a very high accuracy as discussed under Comment 8. The elevation angles of the lidar scans are corrected for this tilt angle in the post-processing phase, hence the observed vertical location of the wake is independent of the tower bending.

Upon the reviewer's recommendation, the authors split the data into the two wind direction sectors (colors corresponding to those used in Fig. 1 of the original manuscript) to test whether the upward displacement is not simply due to topographical effects. The results can be seen in the figure below.

[Figure]

[Figure]

The left figure shows a histogram of the vertical wake center displacement. In both wind direction sectors, indicated by the different colors, the wake center seems to move upwards. This upwards deflection seems to be larger in the sector $\delta < 315°$ (blue) than in the sector $\delta > 315°$ (yellow). However, when looking at the wind speed distributions of these cases (right figure), we see that this sector typically has higher wind speeds. For a fair comparison, we should only consider the same wind speeds. When only considering wind speeds between 5 and 6 m/s (the only bin with approximately the same number of measurements in both sectors), the results look as follows:

[Figure]

The results in this figure suggest that the vertical displacement of the wake center is similar in both sector. However, due to the low data availability it is hard to draw concrete conclusions.

Since these results are not conclusive, as well as that this is out of the scope of what is discussed in this section, only a short notion is added:

"Further analysis (results not shown here) suggests that the vertical wake center displacement, and with that its correlation with input variables, is independent of wind direction. This excludes the influence of topography on these results."

19. Line 431: I am confused here. What are the transparent markers in other panels indicating? I used search function and the word transparent does not show anywhere else. The legend of Fig. 13 also gives no hint.

The authors regret this confusion. Because of the high number of data points, all markers in Fig. 13 are transparent. However, in Fig. 13b (and later in Fig. 13c) the markers appear to be opaque. This is because there are no differences between the resamples, hence there are multiple transparent markers on top of each other, which presents itself as one opaque marker. This is now clarified in the manuscript.

"Additionally, while in Fig. 13e and Fig. 13h transparent markers can be observed, the markers in Fig. 13b appear to be opaque. Here, many transparent markers overlay each other, indicating that GCH estimates the same wake center location in all resamples and that these estimates are not affected by the model's tuning parameters."

20. Line 461: A note should be added here that TI_s serves only as an input for DART and biases are acceptable, because turbulence intensity from a cup anemometer has many problems.

The authors thank the reviewer for this suggestion. The following notion has been added after line 465 of the original manuscript.

"A similar reasoning can be applied to the use of $TI_s$: although turbulence intensity estimates from a nacelle cup anemometer are affected by the rotor (e.g., Barthelmie et al, 2007), biases can be handled by data-driven models and are therefore acceptable."

Technical corrections

1. Line 10: I am not sure that SCADA is a common abbreviation that needs no introduction.

Agreed

2. General: Some abbreviations are introduced multiple times (e.g. DART, GCH). Others are not introduced at all (e.g. NREL, FLORIS).

NREL and FLORIS are now introduced. The full phrase for GCH is now only mentioned in the introduction, while for DART this is done once in the introduction and once in the header of Sect. 3.1 (now Sect. 3.4).

3. Sect. 1-3 in general: I believe a more liberal use of paragraph breaks could be made, when the text moves on to a new thought or new topic.

The authors thank the reviewer for this suggestion. Many new paragraph breaks were inserted.

4. Line 191/192: Maybe: "A flow distortion due to the tower structure affecting the measurements of the met mast occurs for the wind direction sector between X° and Y°, which is not considered in this study (Sect. 2.1). The wind directions analyzed here are assumed to be undisturbed."

Agreed

"A flow distortion due to the tower structure affecting the measurements occurs for wind directions between approximately 310° and 320°, which is not considered in this study (see Sect. 2.1). The wind directions analyzed here are assumed to be undisturbed."

5. Line 251: Maybe: "The same method is applied" instead of "the same thing is done".

This suggestion was adopted.

**Reviewer 2**

General comments

This manuscript contributes to the wind energy field by assessing the quality and performance of a data-driven wake model through a validation experiment with field data.

The authors provide a detailed overview of the measurement campaign and methods used for assessment of the wake models, as well as an extensive consideration of other literature. The results appear to support the conclusion that the presented data-drive wake model outperforms the gaussian-curl hybrid model in terms of prediction of available downstream power. It should be noted that this is only for a downstream distance of four rotor diameters and for a limited range of yaw angles.

Specific comments

1. The authors refer to the potential of the data-driven model as "enormous" and "huge". This appears to be an overstatement in light of the presented results. Suggest to reduce the exaggeration of potential and make more note in the conclusions of the limitations of this data-driven approach.

Although the authors find the results very promising, they acknowledge that more care should be taken when making such exaggerations. Words like "enormous" and "huge" are removed from the manuscript. Statements about the limitations of data-driven models have been added to conclusion and abstract, see Comment 2.

2. The model is claimed to be retrainable, however doing so requires further lidar measurements. Same goes for predictions at other downstream distances. The impact of this requirement on field application needs more emphasis. Additionally, the achieved range of yaw misalignment is considerably smaller that what is used in other literature for wake redirection. It is only briefly noted that the model does not generalise outside of the input range in training data. This limits the potential for application in wake steering control.

The authors agree that the limitation of the data-driven model should be highlighted more. The following has been added to Sect. 5.2:

"DART's quantitative results presented in this study are not fully generalizable. The fitted coefficients in Eq. (3) are only valid for the scenario considered in this study and it is unknown how the model's accuracy transfers to different scenarios, such as other turbine types and downstream distances. Besides, the range of achieved yaw misalignments is typical for field experiments nowadays (e.g., Fleming et al., 2020, 2021; Doekemeijer et al., 2021), although future campaigns could see larger misalignments such as those currently considered in numerical studies (e.g., Howland et al., 2016; Martínez-Tossas et al., 2019; Bastankhah et al., 2022). Further lidar measurements would be needed in new scenarios to guarantee accurate model estimates and although it needs relatively few data to retrained in new situations (Sect. 4.2.4), this limits the potential for application of data-driven models in wake steering control."

The authors were not sure whether the reviewer was referring to other field studies or numerical (or wind tunnel) experiments when they mentioned "other literature". Since other field experiments have also been limited to moderate yaw misalignments, the authors decided to also refer to numerical experiments using yaw misalignments of 30°.

Following Comment 1, statements about the model's limitation have been added to the conclusions and abstract:

"DART shows a high accuracy in the current study, targeting a downstream distance of four rotor diameters and using a range of yaw misalignments commonly used in field experiments. However, these results cannot directly be generalized and further lidar measurements are needed to retrain DART

for new scenarios, limiting its applicability. Regardless, this study's results are believed to demonstrate the potential of data-driven wake models and the role they can play in the further deployment of wake steering control strategies. " (Conclusions)

"Although the results are only obtained for a single turbine type, downstream distance and range of yaw misalignments, the outcome of this study is believed to demonstrate the potential of data-driven wake models." (Abstract)

3. In Section 2.2, concerning the choice of lidar angular velocity and number of PPI scans, it is noted that "too few cases are studied for the statistics to converge". Would that not make the entire comparison invalid? The need to motivate the choice of scanning strategy is clear, but these results appear statistically insignificant?

The authors understand that this statement contradicts the claim in line 120 (original manuscript) to "find the optimum scanning pattern". The goal of this exercise was gaining some insights in how the lidar's scanning pattern affects the quality of the wake reconstruction, rather than to implement a scanning trajectory based on gut feeling. To make statistics converge, many more LES simulations would be needed, representing a wider range of atmospheric conditions and turbine yaw angles. Due to large computational costs, only a pilot with 6 cases (two ABLs and three yaw angles) was performed. Although this is too few for statistics to converge and differences can be attributed to chance, it is believed that these results do allow for making an informed decision.

The following changes have been made:

"In this paper, their strategy was adopted and evaluated numerically to gain insights on how the number of PPI scans and their angular speed (following Carbajo Fuertes and Porté-Agel (2018)) affects the ability to capture the characteristics of 10-minute averaged wake. This exercise used large eddy simulation (LES) results, ..." (line 120)

"...for the statistics to converge. Generating more LES results with a wider range of atmospheric conditions and turbine yaw angles was not possible due to computational restrictions. While these results are not statistically significant and it can therefore not be claimed that an 'optimal' scanning strategy is found, this exercise allows for making an informed decision. It was decided to implement…" (line 148)

Technical corrections
1. Shorter paragraphs would improve structure and readability.

This comment corresponds to technical correction 3 from reviewer 1. The authors thank the reviewer for this suggestion, and many page breaks have been added.

2. Suggest to revise structure, there is quite some inconsistency in length and use of sections / subsections / paragraphs.

The authors have revised the structure and made the following changes:
- Split Sect. 2.1 (Site and experiment) in two sections: 2.1 (Measurement site) and 2.2 (Yaw control experiment)
- Restructure Sect. 3: Previously Sect. 3.1, 3.3, 3.5 are now grouped together in what is now Sect. 3.4. This was done since all these sections relate to DART, whereas previously Sect. 3.2, 3.4 and 3.6 do not. This does, unfortunately, result in three short and one long subsection.
- Split Sect. 3.5 (now Sect. 3.4.4) in unnumbered subsection for DART-3, DART-4 and DART-7
- Split Sect. 4.1 in subsubsections "Correlation with inflow variables" and "Lateral wake center displacement and wake curl"
- Split Sect. 4.2 in subsubsections "Comparison of DART and GCH", "Model accuracy under different conditions" and "Estimating wake characteristics"

3.  Introduction mentions "free-field" experiments, would this not just be a field experiment?

The authors agree that this experiment should be referred to as "field" rather than "free field", acknowledging that there are external influences that affect the collected data. The notion "free" has been removed throughout the manuscript, including title.

4.  Figure 1: colours of topographic map need a colourbar

The copyright statement in the figure needed to be changed before the manuscript was accepted for preprint and the colorbar got lost in the process. It is added again.

5.  Figure 7: black line is not referenced. What is it?

The author regret that this was overlooked. The figure caption now contains the information that the solid lines indicate fitted normal distributions.

6.  Section 4.2: Numerous references are made to the width of normal distribution fits. Consider quantifying by noting standard deviation of fit?

The authors acknowledge that this information could be useful to some readers. However, they are afraid that adding more quantified information could lead to clutter and a harder to read section. Since these values are not essential to support the findings and draw the conclusions, the authors argue to leave this out. However, if the reviewer insists, the authors are willing to add this in a next revision.

7.  Figure 9: indicate the fit quality for the linear fit of trend lines. The associated claim of a "clear" dependency needs more support given the scatter in the data. (ln. 378)

The quality of the fit (expressed by the correlation coefficient R) corresponds to the values in Fig. 8. As requested, these values are now also given in Fig. 9 and referenced in the figure caption:

"The quality of these fits is indicated by the correlation coefficient R, corresponding to Fig. 8."

While making these changes, the authors found a small mistake in the code to generate Fig. 8. This has been fixed now, explaining small differences compared to the figure in the original manuscript.

The authors agree the claim of a "clear" dependency might be too strong considering a correlation coefficient of 0.53. However, it is expected that field measurements contain more noise than numerical simulations, and the authors still believe that these results demonstrate the occurrence of the wake curl in the field.

"Similar to $\mu_y$, the field measurements have a larger spread than the LES results, expressed by the lower quality of the linear fit (correlation coefficient R). However, the fitted lines are similar, indicating that the wake curl does indeed occur in field, something that until now had not conclusively been shown in literature."

**Reviewer 3**

General comments

The manuscript by Sengers et al. presents the (free-field experimental) validation of the DART model which was introduced by the authors before. Model predictions with a focus on wake steering of a single turbine are compared with both the experimental data, comprising measurements of two scanning lidars and the turbine data, and the Gaussian-Curl Hybrid (GDH) model as reference. Overall, the manuscript meets its defined objectives and describes the findings in a well comprehensible manner. However, I have identified three main deficits, I will briefly summarize (as major issues), followed by a list of minor points which should be addressed before a publication of the manuscript.

Major issues:

1. As I understand it, the manuscript contrasts from the earlier publication(s) by using free-field data for validation. The used measurement campaign has however, as stated by the authors, several shortcomings. These are described but their impact on the results is, in my view, not sufficiently discussed and quantified, respectively. A more detailed uncertainty quantification, elaborating on these impacts, may be very useful to underline the findings of the study.

Many of the sources of uncertainty mentioned in the manuscript (e.g., lidar accuracy and scanning strategy, filtering, impact of vegetation) are unavoidable in field measurements and were already mentioned in Sect. 5.1. As noted by Reviewer 2, technical correction 3, this campaign should be described as "field" campaign, rather than a "free-field" campaign, acknowledging that there are external influences that affect the collected data. However, from the analysis performed in this study, specifically referring to Sect. 4.1, the measurements appear to be physically reasonable.
This uncertainty can only properly be addressed by carrying out a multitude of measurement campaign, varying the site, turbine type and device settings. Besides, it should be stressed that it is not claimed that the DART model has a certain error, but rather that it has this error on this specific data set. This has been addressed under Reviewer 2, comment 2, on the limitations (more specifically the generalizability) of the data-driven model.

However, on the reviewer's request, the authors have carried out an uncertainty analysis, tackling in their eyes the biggest sources of uncertainty in this study. This includes the lack of a hard target analysis of the nacelle-mounted lidar, as well as the measurement accuracy of the met mast devices. The following was added to the manuscript as subsection 5.1.2:

[revised manuscript text omitted]

Although these tests do only allow for systematic biases, it gives an idea of the robustness of the wake models to measurement errors. A more detailed analysis, for instance adding a PDF of errors to the measurement rather than one value, is deemed outside of the scope of the current work.

2. Major issue #2: The main findings of the presented study focus on the accuracy of wake characteristics and the comparison to results of the reference model. What I am missing is a

more detailed discussed how these findings impact the possible application of wake steering in terms of the introduced control strategies in wind farms. I suggest to add another sub-section in the Discussion (section 5) which addresses this, the implications (qualitative and quantitative) on an application. In addition to seeing the need for this discussion, I also think the current section 5 is rather short and should be elaborated on.

The authors thank the reviewer for this comment and fully agree that a section on the implications for application is missing. The following has been added to Sect. 5.

"**5.3 Implications for future work**

As noted in the introduction, the industry appears to be hesitant to adopt the wake steering strategy due to large uncertainties. To overcome this, yaw controllers need to become more sophisticated, for instance by using closed-loop controllers (Doekemeijer et al.,2020; Howland et al., 2020) or using preview information (Simley et al., 2021b). On the other hand, the low-fidelity wake models that are utilized to determine the yaw misalignment set points used by the yaw controller need to become more accurate.

This study contributes to the latter by showing that both DART and GCH perform well on average (small systematic bias), but that DART can capture a higher degree of variability observed in the field. Besides more accurate estimations of the wake deficit, which historically has been the main focus of wake models, this extends to other wake characteristics like wake curl and wake center location. The latter is especially important for wake steering, as erroneous steering can steer the wake into a downstream turbine.

Since DART shows a higher accuracy than GCH in estimating wake characteristics, it can be hypothesized that when using DART to determine yaw misalignment set points used by the yaw controller, the wake steering strategy can be applied more successfully. This can consist of achieving higher power gains when wake steering is performed successfully, or reducing power losses due to erroneous steering. However, an extensive campaign would be needed to investigate this, which was considered out of the scope of the current work.

On a more general level, this study shows that data-driven models are a viable alternative to analytical models. Whereas data-driven models have often been criticized for their complex nature, this study has demonstrated that accurate estimations can also be obtained with a very simple linear model.

While the current model focuses on estimated wind speed and consequently power, a similar methodology could be developed to estimate turbulence and consequently turbine loads. Alternatively, it would be interesting to combine analytical and data-driven models in hybrid models. Such models could initially benefit from the robustness of analytical models, but exploit the higher accuracy of data-driven models when more data becomes available."

The reviewer suggested to describe these implications also quantitatively. However, it is impossible to directly quantify the influence of this new wake model on the effectiveness of wake steering (its eventual application) without additional measurements. It has been clarified that this is considered out of the scope of the current work.

In addition, Sect. 5 has been extended following Comment 1 and Reviewer 2, Comment 2.

3. Major issue #3: Though clearly written, I think the presentation and first of all the structure of the manuscript should be improved. Section order / levels and titles should be revised and in some cases made more consistent. More concrete suggestions in the list below.

The authors want to thank the reviewer for their concrete suggestions. This was very helpful when revising the structure of the manuscript. Details about what was changed and added are given below at their respective suggestion.

Minor issues:
1. l. 26 – "test turbines" instead of "testing turbines"

Agreed

2. l. 27 – Referring to "simulations and wind tunnel experiments", I think it would be helpful to shortly describe in some more detail related pros and cons (of these) in contrast to a free-field experiment.

The following was added:

"Although they provide higher degree of reproducibility and more flexibility in choosing the studied scenarios, these experiments take place in controlled environments and do not fully represent the complexity of the field."

ll. 29 – Please describe in one more sentence how "erroneous yawing" may interact with wake steering more explicitly.

The phrase "erroneous yawing" in this sentence was perhaps too premature, especially since this was not directly used in the provided reference. This sentence was rewritten in a more general way, and an additional reference was added.

"This uncertainty is amplified by findings that the application of wake steering can lead to power losses under certain conditions (e.g., Fleming et al., 2020; Doekemeijer et al., 2021)."

The phrase "erroneous yawing" is still used in lines 44-45 of the original manuscript, including what this indicates for wake steering.

3. ll. 73 – Do we need explicit wake steering for such an experiment or is it just about the variation of yaw angles? Please comment on this.

In the current context, only a variation of yaw misalignment is required.

"As mentioned before, studies validating wake models with field measurements are rare, especially when yaw misalignments are included, resulting in uncertainties about their accuracy."

4. ll. 75 – How do you define the difference between "validations and comparisons"? Please comment.

This has been clarified.

"However, validations with measurements and comparisons between models are necessary to assess their performance and provide direction for future work."

5. l. 79 – What does "This" refer to?

"This" refers to the overall objective of this study.

"To achieve the objective, this study comprises of three components:"

6. ll. 79 – I suggest to use commas for this list.

Agreed

7. l. 84 – Please write a short introduction to this section, also introducing the sub-sections.

The authors agree that this section needs an introduction. This has been added.

"This section introduces the field experiment carried out within this study. Section 2.1 describes the measurement site and general setup. Section 2.2 describes the yaw control experiment. Sections 2.3 through 2.7 then discuss the devices, their measurement strategies and data processing. Especially in Sect. 2.3 more details are provided, including results from a preliminary study to determine the scanning strategy of the nacelle lidar, since the measurements from this device are essential for this study. Lastly, Sect. 2.8 describes how the data from all devices are used to select 10-minute averaged cases considered in the rest of the study."

8. ll. 94/95 – Can you elaborate on this, why only a fixed yaw offset in this sector?

This sentence has been rephrased.

"As these measurements were part of a larger field campaign, only the wind direction sector $\delta$ = [268 °, 360 °] ∪ [0 °, 20 °] could be used for experiments for this study."

Besides, a short notion was added to the end of the section explaining why only fixed yaw offsets were applied.

"Fixed yaw offsets were applied as this involved minimal changes to the yaw controller. Besides, a distribution of yaw misalignments was expected to be obtained due to the imperfect tracking of the wind direction by the yaw controller."

9. ll. 112 – I do not think this title fits optimally to the other titles. Again, I also suggest to add a short introduction to the following sub-sub-sections. And why do you not use numbering (2.2.1, ..) for the following paragraphs?

The header of this section was changed to make it more similar to the other header of this section. Numbering of subsubsections was added. A short introduction is now provided:

"This section describes the measurements performed with the nacelle-mounted lidar. Section 2.3.1 describes the design of the scanning strategy, including results of a numerical evaluation to determine what trajectory should be implemented in the field. Section 2.3.2 describes the processing, including filtering, of this data."

10. ll. 114 – This sentence ("A pulsed ..") needs to be rewritten. Please check again what you want to say.

This sentence has been rewritten.

"A pulsed lidar can be mounted onto the nacelle to sample to turbine's wake. When operated with a single plan position indicator (PPI) scan with an elevation angle of 0°, the line-of-sight velocities on a horizontal plane at hub height are obtained. Although quick, this trajectory only provides data at one height in the wake. "

11. l. 115 – "horizontal plane" is only true if you use a single PPI with zero elevation – please be more explicit here.

Resolved with Comment 11.

12. Figure 1 – The colour scheme is not optimal here (in particular, as you are not showing the colour bar as legend). Could you use less blue?

The blue has been replaced by red in Fig. 1 and the color bar was added (see Reviewer 1, technical correction 4). Correspondingly, Fig. 4 was updated.

13. l. 137 – Please detail the link between U_eq and P_av.

The available power ($P_{av}$) represents the energy available in the flow, not considering turbine efficiency (e.g., $C_P$). Equation (1) now only includes the first part, and the following is added directly after:

"… in which $P_{av} = P / C_P = 0.5 \, \rho \, A \, U_{eq}^3$ with $\rho$ the air density (assumed to be constant), $A$ the rotor area $U_{eq}$ the rotor equivalent wind speed."

14. l. 138 – Delete one "with".

Done

15. l. 139 – Why do you give the equation for PE but not APE here? This is rather confusing as you write about APE before.

This was originally done because in line 322 a PE is calculated and referred to this equation. However, the authors acknowledge that this is confusing here. Eq (1) now calculates the APE, while in line 322 the following has been added:
"…calculated analogous to Eq. (1), but without absolute values…"

16. l. 140 – "reconstruction" of what?
"…the reconstruction of the wake…"

17. l. 142 – ".. were not tested, as this would remove .." This is only true if you require a symmetry. Please comment on this.
Correct, however a symmetry is desirable for fitting Gaussian or quadratic equations through the data points (see Sect. 2.8).

18. l. 145 – "hold" instead of "holds"
Corrected

19. Figure 3 – In my print it is not really "yellow" – please check, and maybe use another colour.
A brighter yellow is now used.

20. Figure 3 – Why was the "second cluster [..] omitted"? I think this is neither sufficiently described in the figure caption nor in the main text.
The author agree that the main text should explain why this cluster is omitted. The following has been added:
"Clusters were then either considered or eliminated based on whether the location of their center was physically feasible. In the example in Fig. 3b the yellow cluster was omitted, since many points outside the main cluster with high CNR and low LOS values indicate erroneous measurements."
The occurrence of this second cluster could be due to a misinterpretation of the peak finder in the spectrum. However, analyzing this was considered out of the scope of the current work.

21. l. 167 – Please explain briefly why you have "slightly different azimuth angles".
This is due to the nature of the scanning device. Especially since in this study the scanning trajectory is very fast, not all measurement points are exactly the same.
"...to account for the slightly different azimuth angles between scans as a result of the lidar's inability to measure the exact same location each time."

22. l. 174 – Please mention here again that the ground-based lidar is also a scanning lidar of type 200s.
The following has been added to Sect. 2.1
"A second pulsed lidar of the same type (serial no. WLS200S-023) was installed west of the turbine to measure inflow profiles (VAD, Sect. 2.4)."
The authors have the preference to mention all devices names and brands in Sect. 2.1 and only discuss their usage in subsequent sections. This is also done for the other devices.

23. l. 187 – Here you could introduce the abbreviation "met".
"A meteorological (met) mast…"

24. Figure 5 (and others) – Difference symbols are used for veer in figures and text – "del" and "delta", respectively.
This is corrected.

25. l. 239 – Again, I suggest to add some introduction text to the section.
The following was added.
"This section introduces the modeling aspects of this study. First, This section introduces the modeling aspects of this study. First, Sect. 3.1 summarizes the Multiple 1D Gaussian method used to obtain quantifiable wake characteristics. Sect. 3.2 discusses what information is used as a reference and Sect. 3.3 describes the splitting of the data set in training and testing subsets. Then, Sect. 3.4 introduces the data-driven model and Sect. 3.5 briefly introduces the analytical model used in this study."

26. l. 251 – "their" instead of "it's"
Agreed

27. l. 278 – There is only one sub-level (3.1.1) – please revise section structure.
This was already adjusted following Reviewer 2, Technical correction 2.  Some introduction text was added to what is now Sect. 3.4:
"This section introduces DART, starting with a summary from previous work in Sect. 3.4.1 and changed made to the model since this work in Sect. 3.4.2. This is followed by information on the input variables (Sect. 3.4.3). Lastly, the feature selection of the three version of the model considering in this study is discussed in Sect. 3.4.4."

28. l. 300 – I believe this should say "for this particular experiment" rather than "in the free field".
"...are weakly correlated in this field experiment."

29. Figure 8 – As above, use "delta alpha" as in main text.
This is corrected.

30. Figure 10 – Please introduce the colour code for (a).
The following was added to the figure's caption:
"The colorbar applies to both figures."

31. Figure 11 – The "yellow" text is very difficult to read – please select another colour.
Due to readability for readers with color vision deficiencies, the authors are restricted in choosing the colors used in the figures. However, the text is now printed in a slightly darker yellow, which the authors believe increases visibility. The same is done for the text in Fig. 13.

32. l. 437 – There should be some introduction to the following sub-sub-sections.
Introductions have been added to Sect. 4.2 (referred to in this comment) as well as to Sect. 4 and Sect. 4.1.
"**4 Results**
This section presents the results of this study. Section 4.1 describes the characteristics of the wake observed in the field, after which in Sect. 4.2 the performance of the wake models in reproducing these wake characteristics is discussed.

**4.1 Observed wake characteristics**
In Sect. 4.1.1 an assessment of the characteristics of the observed wake listed in Table 1 is performed, which is deemed a necessary first step before investigating the accuracy of wake models. The observed wake characteristics are linked to the inflow variables to examine whether the measurements are physically feasible. In Sect. 4.1.2, two wake characteristics that are deemed important for wake steering are further investigated.

**4.2 Performance of wake models**
This section presents the performance of the DART and GCH wake models in reproducing the wake characteristics observed in the field. Section 4.2.1 presents how well the models can reproduce the available power measured by the lidar. Following this general result, Sect. 4.2.2 zooms in on how well the model perform under different conditions and Sect. 4.2.3 displays how well the models can reproduce a selection of wake characteristics. Section 4.2.4 discusses how sensitive the models are to the amount of training data. Lastly, Sect. 4.2.5 evaluates how well DART performs when only using SCADA data as input."

33. l. 444 – "20%" of the original dataset – suggest to add this detail here.
Agreed

34. l. 480 – Add some introduction to the section here.
The following introduction was added:
"Section 5.1 discusses the measurement campaign and its accuracy. In Sect. 5.2 the limitations of the data-driven model are reviewed. Finally, Sect. 5.3 focuses on the implication of this study's results for future work."

35. ll. 480 – As pointed out above, this section should be elaborated on. Currently it only addresses the limitation. I suggest a more in-depth discussion of the application here.
See discussion Main Issue 2

36. Appendix A – I do not think this appendix is really needed.
The authors do not have a strong opinion about this and would leave this decision to the editor.